# Translating Flowering Time from *Arabidopsis thaliana* to Brassicaceae and Asteraceae Crop Species

**DOI:** 10.3390/plants7040111

**Published:** 2018-12-16

**Authors:** Willeke Leijten, Ronald Koes, Ilja Roobeek, Giovanna Frugis

**Affiliations:** 1ENZA Zaden Research & Development B.V., Haling 1E, 1602 DB Enkhuizen, The Netherlands; W.Leijten@enzazaden.nl; 2Swammerdam Institute for Life Sciences (SILS), University of Amsterdam, Science Park 904, 1098 XH Amsterdam, The Netherlands; r.e.koes@uva.nl; 3Istituto di Biologia e Biotecnologia Agraria (IBBA), Operative Unit of Rome, Consiglio Nazionale delle Ricerche (CNR), via Salaria Km. 29,300, 00015 Monterotondo Scalo, Roma, Italy

**Keywords:** Brassicaceae, Asteraceae, flowering time, photoperiod, vernalization, ambient temperature, gibberellins, age, plant breeding

## Abstract

Flowering and seed set are essential for plant species to survive, hence plants need to adapt to highly variable environments to flower in the most favorable conditions. Endogenous cues such as plant age and hormones coordinate with the environmental cues like temperature and day length to determine optimal time for the transition from vegetative to reproductive growth. In a breeding context, controlling flowering time would help to speed up the production of new hybrids and produce high yield throughout the year. The flowering time genetic network is extensively studied in the plant model species *Arabidopsis thaliana*, however this knowledge is still limited in most crops. This article reviews evidence of conservation and divergence of flowering time regulation in *A. thaliana* with its related crop species in the Brassicaceae and with more distant vegetable crops within the Asteraceae family. Despite the overall conservation of most flowering time pathways in these families, many genes controlling this trait remain elusive, and the function of most Arabidopsis homologs in these crops are yet to be determined. However, the knowledge gathered so far in both model and crop species can be already exploited in vegetable crop breeding for flowering time control.

## 1. Introduction

The switch from vegetative stage to flowering is essential for plant reproduction, and flowering time diversity has adaptive value in natural populations [1]. The time at which flowering occurs plays a major role in agricultural production as it affects the quality and quantity of leaf, flower, seed and fruit products, ease of harvest and marketing. Shifting the seasonal timing of reproduction is a major goal of plant breeders to develop novel varieties that are better adapted to local environments and changing climatic conditions [2]. Over the last few years, climate underwent significant changes, such as relatively mild winters, dry and warm summers, and more heavy rain fall in spring and autumn. All of those changes affect plant growth and flowering time. Besides natural occurring climate change, adapting varieties to new environments makes crop production more flexible [2]. To produce varieties that are more robust and predictable in flowering time is also a desirable trait for reliable production. Obtaining varieties with increased yield is also a major breeding goal, and will enhance food production within the same amount of land in a world where the population is growing, and demanding more food production. However, yield is influenced by several factors, including premature bolting (see Glossary in Table 1) during crop production, and therefore, a prolonged vegetative phase will increase yield for leafy crops that are harvested before the transition to the reproductive phase. 

Controlling flowering time would therefore help grow crops in all seasons to speed up the production of new hybrids and produce high yield throughout the year. Early bolting potentially limits vegetative growth and can severely decrease yield, while non-flowering inhibits seed production. The timing of bolting and flowering are especially important for vegetable crops. For cauliflower and broccoli, synchronization of flowering is essential as the plants are harvested in the inflorescence meristems phase (curds). For lettuce, plants flower early when grown at high temperature. Early stages of bolting are not visible, but the flavor changes more towards bitter. Therefore, late bolting is preferred to enhance yield without the bitterness.

In the past, selection for flowering time was based on plant phenotyping in the greenhouse or in the field. The increasing availability of crop genetic and genomic resources, and the current knowledge on both gene function and natural genetic variation, are of great value and can be used in breeding. The development of trait specific markers, e.g., based on QTL (see Glossary in Table 1) analysis, are useful to select for favorable genotypes in a breeding program [3,4,5]. On the other hand, reverse screening for genetic variation in specific flowering time related genes in wild accessions or mutant populations could be of benefit for the trait. The latter approach is still underexploited as knowledge about the flowering pathways, the molecular mechanisms, and the genes involved is still limited in most crops [6,7]. However, the flowering time genetic network is extensively studied in the model species *Arabidopsis thaliana* [6,8,9,10,11] (Figure 1a), which is an annual facultative long-day (LD) plant belonging to the Brassicaceae family. Hence, if the function of the *A. thaliana* flowering time genes would be conserved in the crops of interest, this would provide targets for genetic selection and improvement to speed up breeding and agricultural biotechnology.

Here, we review current knowledge regarding the conservation and divergence of the mechanisms that regulate flowering time in *A. thaliana* related crop species from the Brassicaceae family and more distantly related leafy crops within the Asteraceae family, that are of great interest for food market and vegetable breeding. We will focus on *Brassica napus* (Figure 1b), *Brassica rapa* (Figure 1c), *Brassica oleracea* (Figure 1g-h), *Raphanus sativus* (Figure 1d and *Diplotaxis tenuifolia* (Figure 1e) for the Brassicaceae and on *Lactuca sativa* (Figure 1i) and *Cichorium intybus* (Figure 1f) as key leafy crops within the Asteraceae family. The possible exploitation of this knowledge in vegetable crop breeding, and the potential of translational biology and genomics to crops, will be discussed. The different flowering time pathways and genes explored in these crops will be discussed, and gene function will be compared to the knowledge acquired in *A. thaliana*.

## 2. Flowering Requirements of Brassicaceae and Asteraceae Species

Optimal conditions for flowering vary between and within species, as plants respond and adapt to specific combination of light (quality and day length) and temperature (cold, warm, hot) to undergo floral transition [11]. Plants can be either long-day or short-day if they flower when exposed to light periods longer (as in summer) or shorter (as in winter) of a certain critical length. Many plant species require prolonged exposure to low temperatures (vernalization) to flower, while others flower independently of cold conditions [8]. Plants have annual, biennial or perennial life cycles depending on the number of growing seasons required to complete their life cycle [11]. Further classification of plant types can be made based on geographical origin and growing season. 

The plant model species *Arabidopsis thaliana* (Figure 1a, Table 2) can be annual or biennial. Annual plants flowers earlier in response to long days (facultative long-day), and natural accessions are classified into summer annuals and winter annuals [11]. Summer annuals flower rapidly when grown under long days, whereas in winter annuals flowering is not induced until they are exposed to low temperature (4 °C) for several weeks (vernalization), followed by warmer temperatures in spring [11].

*B. napus*, *B. rapa* and *B. oleracea,* closely related to *A. thaliana*, share similar life cycles (annuals and biennials) and have spring, semi-winter and winter types: Spring types flower early without vernalization and are grown in geographical regions with strong winters or in subtropical climates; winter types have an obligate requirement for prolonged periods of cold temperatures and are grown in moderate temperate climates; semi-winter types, which are sown before winter, flower after winter and are grown in geographical regions with moderate winter temperatures (>0 °C) (Figure 2) [14]. *B. napus* is a domesticated allotetraploid species with two genomes, AA and CC, derived from *B. rapa* and *B. oleracea*, respectively. Two other Brassicaceae species, *R. sativus* and *D. tenuifolia*, and the Asteraceae species *Lactuca sativa,* are annuals that do not require vernalization to flower. In contrast, the Asteraceae species *Cichorium intybus* is biennial or perennial and does require a cold period for flowering induction. All species considered in this review flower faster in response to longer photoperiods and warmer temperatures (more details about species characteristics are in Table 2). Despite similarities amongst these species, breeding goals with regard to plant growth and flowering time differ for each crop.

## 3. Breeding Goals

Breeders aim to improve varieties by adapting them to climate changes, new environments or increasing yield in general, and flowering time affects all these traits [2].

*B. napus* L. (Figure 1b) is one of the most important oilseed crops worldwide and includes oilseed rape and rapeseed (Figure 3). The yield potential of rapeseed largely depends on flowering time, thus creating lines with optimal flowering time is a major breeding goal [14].

*B. rapa* (Figure 1c) is cultivated worldwide, particularly in Asia, and includes the vegetable crops Chinese cabbage, pak choi, turnip and cime di rapa (Figure 3). Premature bolting is a severe problem as it reduces yield of the harvested crops, e.g., for the spring cultivation of Chinese cabbage. Extremely late bolting is a major breeding goal in this crop as unexpectedly low temperatures can induce flowering and so yield loss [23].

*B. oleracea* (Figure 1g–h) encompasses multiple cultivar groups (Figure 3) that are classified based on the morphology of their edible structures: Kohlrabi, kales and cabbages are harvested at vegetative stage; broccoli and cauliflower are cultivated for their curd (the edible flower head of the plant) that is harvested at the transition to reproductive phase. Cultivars and wild species accumulate anti-carcinogenic and antioxidants, which are beneficial for human health [16]. Breeding strategies for broccoli and cauliflower include uniformity in time to curd production for easy crop handling during production. In cauliflower, slight deviations from optimal growth temperature, either lower or higher, lead to uneven curd formation and therefore less predictable harvest times. On one hand, vernalization is required to produce a harvestable curd, while on the other hand, high temperatures in spring result in prolonged vegetative growth before the curd is produced. Adjusting the vernalization and temperature sensitivity of plants will help to create cauliflower varieties with a predictable curd formation, for example by exploring genetic variation in temperature-dependent flowering time genes. Early prediction of the thermal time to curd induction in untested genotypes and environments can be achieved by using the genome-based model proposed by Rosen et al. [24], making this a good tool for early selection of the desired genotypes to be incorporated into breeding material.

*R. sativus* (Figure 1d), including radish and daikon (Figure 3), are important vegetable root crops with large variation in root size and shapes [18,25]. Late flowering is a relevant breeding goal as some varieties are sensitive to premature bolting. In radish, when plants are grown under LD conditions, premature bolting reduces yield and quality of the harvested product. Enhancing the quality of radish can be achieved by late flowering, combined with increased post-harvest shelf life through delayed leaf senescence, as the whole plant (bulb and leaves) is harvested and the leaves are used as an indication for the post-harvest quality of the plant.

Wild rocket (*D. tenuifolia*) (Figure 1e and Figure 3) is a popular salad leaf that has a similar shape and taste as rucola (*Eruca sativa*), but with a stronger peppery flavor. The crop can be harvested multiple times. For cultivation, *D. tenuifolia* is selected against pre-harvest flowering as it leads to unsaleable crops.

*L. sativa* (Figure 1i) encompasses multiple lettuce cultivars that are classified based on the morphology of the leafy or head type: Iceberg types form a close head resembling that of a cabbage; butterhead types form a head with large ruffled outer leafs; and romaine (Cos) types do not form a head but have long, broad and upright leaves. Wild relatives used in breeding include *L. virosa*, *L. serriola* and *L. saligna* (Figure 3). While cultivated lettuce is annual, the wild species *L. virosa* is biennial and does require vernalization for flower initiation. High temperature during the cultivation of lettuce results in heat-induced early bolting. Heat resistance is therefore a major breeding goal to produce better tasting lettuce when grown at high temperature. Although early bolting is beneficial for fast seed production in the creation of varieties, it does reduce the quality of harvestable crops. The early stages of bolting are not visible when the crop is harvested, but the flavor changes towards an undesirable bitter taste [26]. Exploring the genetic determinants of this response will help understand the mechanism of heat-induced early flowering, and enable breeders to produce better tasting lettuce when grown at high temperature.

*C. intybus* (Figure 1f) includes multiple cultivar groups that are classified based on purpose and use of the harvested product: Root and forage chicory is used for inulin extraction and grown for live stock, respectively; witloof and radicchio are leafy vegetables that can be cooked or eaten fresh [27]. Among the leaf chicory groups, several “Catalogna” landraces are cultivated in Italy for both leaves and stems/buds, the latter appreciated for the bitter and crispy taste (puntarelle) (Figure 3) [28]. If sown too early in spring, the plants could be vernalized and flower during the first growing period [22]. Breeding goals include uniformity in crop yield and maturity, and resistance to bolting [22,29]. For the production of root chicory, a cold season during growth induces early bolting and therefore decreases yield. Investigating the cold-response of root chicory in more detail is needed to delay bolting under these conditions.

## 4. Conserved and Divergent Flowering Time Genes in Brassicaceae and Asteraceae

The switch from plant vegetative growth to reproductive development (transition to flowering), is a critical stage in the life cycle of a plant. Plants need to coordinate their developmental programs precisely in response to seasonal changes and in an ecological context in order to ensure their reproductive success. As such, flowering is tightly controlled by diverse developmental, hormonal and environmental cues, day length and temperature being the most important of these environmental signals. [10]. Six major genetic pathways, converging to a small number of floral integrator genes, have been described in *Arabidopsis thaliana* (Figure 4): The vernalization and photoperiod pathways, which control flowering in response to seasonal changes; the ambient temperature pathway, which regulates flowering time in response to changing ambient temperature; the age, autonomous, and gibberellin pathways, acting more independently of environmental stimuli [9,10]. When the switch towards flowering is made, the shoot apical meristem (SAM) transforms into an inflorescence meristem (IM) as an intermediate step. From the IM, floral meristems (FM) are initiated that can produce floral primordia [30]. The transition to reproduction is accompanied by shoot stem elongation (bolting).

### 4.1. Floral Integrator Genes: An Overview

In plants, the signaling pathways that are activated by various endogenous and environmental cues ultimately converge to a few floral integrator genes to control flowering time, leading to the activation of floral meristem identity genes, the first step in the formation of a flower [10]. In Arabidopsis, two floral integrators play a major role in the transition to flowering, FLOWERING LOCUS T (FT), which belongs to the PEBP (phosphatidylethanolamine-binding protein) family, and SUPPRESSOR OF OVEREXPRESSION OF CONSTANS 1 (SOC1/AGL20), a MADS-box transcription factor [9] (Figure 4). Two homologs of FT, TWIN SISTER OF FT (TSF) and TERMINAL FLOWER 1 (TFL1), act redundantly or antagonistically to *FT*, respectively [31,32]. If floral integrator genes were conserved between Arabidopsis and crops, mutations reducing *FT*, *TSF* and *SOC1* orthologs activity would result in late flowering plants, whereas increased expression of the corresponding genes should induce early flowering. The opposite would occur for mutations or increased expression of *TFL1*.

The *FT*/*TFL* family and *SOC1* have a major role in flowering time response that seems to be conserved across different species [34]. However, due to Brassica genus evolutionary history, with genome triplication within diploid species of Brassica and polyploidism, *B. napus*, *B. oleracea* and *B. napus* contain several copies of floral integrator homologs. Among these, only some conserve a key role in flowering time whereas others may have been either inactivated or undergone a process of neofunctionalization [35]. Lettuce *SOC1* shows a unique role in heat-promoted bolting [36], whereas its role downstream of *FT* to induce flowering transition in natural conditions is yet to be determined.

#### 4.1.1. Floral Integrator Genes in Brassicaceae

In Arabidopsis, FT plays a key role in the floral transition process, since it is the mobile signal moving from the leaves, through the phloem, to promote flowering at the SAM [37,38]. In the leaves, the circadian clock-associated gene *CONSTANS* (*CO*) enhances *FT* expression under long-day (LD) conditions, while temperature-dependent genes such as *SHORT VEGETATIVE PHASE* (*SVP*) and *FLOWERING LOCUS C* (*FLC*) repress *FT* expression independently of day length [39,40,41]. FT protein is produced in the leaves and transported through the phloem to the SAM, with FT-INTERACTING PROTEIN 1 (FTIP1) assisting with FT protein transport [37]. At the SAM, FT interacts via a 14-3-3 protein with FLOWERING LOCUS D (FD) [42] to activate the floral promoter *SOC1* and the downstream floral meristem identity genes *APETALA 1* (*AP1*) and the AP1 paralog *CAULIFLOWER* (*CAL*) [38]. SOC1 also activates floral meristem identity through *LEAFY* (*LFY*) and *AGAMOUS-LIKE 24* (*AGL24*). LFY, AP1 and CAL direct certain groups of cells in the flanks of the SAM to differentiate into floral meristems, leading to the transition from vegetative to reproductive phase [43]. A close homolog of *FT*, *TWIN SISTER OF FT* (*TSF*) with 82% amino acid identity, seems to act redundantly to *FT*. Overexpression of either *FT* or *TSF* results in early flowering, a mutation in *FT* results in late flowering under LD conditions and a mutation in *TSF* shows a greater effect under short-day (SD) conditions [31]. A more distantly related homolog of *FT*, *TERMINAL FLOWER 1* (*TFL1*) with 59% amino acid identity, acts antagonistically to *FT*. Plants overexpressing *TFL1* are late flowering with an extended first inflorescence phase, during which they form cauline leaves and branches [44]. TFL1 is a mobile signal like FT [45], but acts as a transcription repressor rather than a transcriptional activator as FT [46]. The antagonistic activity of TFL1 and FT originates from an external loop in the protein [32] and interchanging one specific residue in the loop (Y85 in FT and H88 in TFL1) is sufficient to convert TFL1 into FT function and vice versa [44].

*B. napus* contains six paralogs of *FT*, four of *TFL1* [47,48] and four paralogs of *SOC1* (Table 3) [35]. *FT* paralogs map to six distinct regions of conserved blocks of the A and C genomes homologous to a common ancestral block of *Arabidopsis* chromosome 1. *BnFT* gene coding sequences show 92% to 99% identities to each other and 85% to 87% identity with those of Arabidopsis. *Bna.FT.C02* and the corresponding ortholog in *B. oleracea* are not expressed, possibly due to the presence of a transposable element (TE) causing high cytosine methylation at the promoter [47]. Differently, three *Bna.FT* paralogs, *Bna.FT.A02, BnaFT.C06a* and *Bna.FT.C06b*, are expressed and were associated with two major QTL clusters for flowering time, one of which encompasses two *Bna.FT* paralogs *Bna.FT-C06a* and *Bna.FT.C06b*. Their function in flowering time variation was confirmed by association analysis in vernalization-free conditions in both spring and winter type cultivars of rapeseed. *Bna.FT.A02* is expressed in leaves of both winter and spring type plants, with and without vernalization [47,49], and was found to associate with flowering time in a panel of 188 *Brassica* spp. accessions collected from different geographic locations worldwide [50]. *Bna.FT.C06* and *Bna.FT.A07* are expressed in winter type plants after vernalization and spring type plants, but not in winter types without vernalization [47,49]. EMS (ethyl methanesulfonate) lines harboring different mutant alleles of *Bna.FT.C06b* were late flowering and displayed reduced fertility [51]. Plants harboring different mutant alleles of *Bna.TFL1* paralogs were not affected in flowering time [51], whereas amongst the four paralogs of *SOC1*, only *Bna.SOC1.A03* was associated with flowering time and seed yield-related QTLs on chromosome A03, and its expression was induced by vernalization [52]. These data point to a function of *Bna.FT.A02* and *Bna.FT.C06* in controlling flowering time, the latter also in response to vernalization similarly to their paralog in Arabidopsis, whereas *B. napus* paralogs of *TFL1* seem to affect seed yield but not flowering time [51]. *Bna.SOC1.A03* might play a role in flowering time control, but this is yet to be explored in *B. napus* species. These data are consistent with the association of *Bna.FT-A02,* but not *B. napus TLF1* and *SOC1* paralogs, with a spring-environment specific flowering QTL in double haploid populations grown in different environmental conditions [53].

Three paralogs of *SOC1* (*Br004928, Br000393* and *Br009324*) and two paralogs of *FT* (*BrFT*) are found and expressed in *B. rapa* (Table 3) [47,54]. *BrFT1* and *BrFT2* show a similar expression as their corresponding *B. napus* orthologs *Bna.FT.A02* and *Bna.FT.A07*, respectively. *BrFT1* is expressed in all plant types and diurnally regulated [47,54]. *BrFT2* is only expressed in winter type plants after vernalization and spring-type plants [47]. A TE in exon 2 of *BrFT2* causes plants to flower 4.9 days later in spring and 14.7 days later in autumn. Due to the bigger effect under SD conditions, it was suggested that *BrFT2* might be an ortholog of *AtTSF* [49], but this still needs to be confirmed. Overexpression of a *B. rapa SOC1* ortholog (*BrAGL20*) in *B. napus* causes early flowering [55], suggesting that the function of this gene may be conserved in Brassicaceae. Moreover, association between flowering time and expression of the two *SOC1* paralogs *Br004928* and *Br000393* was found in a natural population of *B. rapa* [56].

Four paralogs of *FT* (*BoFT*) (two copies on C02 and one on C04 and C06), and three homologs of *SOC1* (*BoSOC1*) (C03 and two copies on C04) have been identified in the genomes of *B. oleracea* (Table 3) [35], but no functional studies are available so far.

One *FT*, *TFL1*, *TSF* and two *SOC1* genes, sharing 82.58%, 89.47%, 83.3%, 85.49% and 88.82% of nucleotide homology with their Arabidopsis homologs, can be found in the de novo assembled transcriptome of *D. tenuifolia* [57] that was obtained from leaves of stressed young plants. However, no characterization of floral integrator genes is available for this species.

#### 4.1.2. Floral Integrator Genes in Asteraceae

In *L. sativa*, an *FT* homolog (*LsFT*) was characterized [58] and shown to express in the largest lettuce leaves, stems and flower bud in controlled high temperature (35/25 °C) conditions which induce lettuce flowering [58]. *LsFT* overexpression could induce early flowering in transgenic *A. thaliana*, although the phenotype was less strong compared to *AtFT* overexpressing plants. However, other studies showed that expression of *LsFT* under the viral 35S constitutive promoter control could fully complement *Arabidopsis ft* null mutant [59]. Correlation between *LsFT* expression and lettuce bolting (measured as the days to the first visible elongated stem) was further analyzed in nine lettuce varieties, which were selected amongst 705 lettuce accessions, with either late, middle and early bolting times [59]. Heat treatment (35 °C day/ 25 °C night) for 48 h also promoted expression of *LsFT* in all lettuce varieties. RNAi-mediated knockdown of *LsFT* in *L. sativa* results in a late bolting phenotype, lack of response to heat treatment and reduced levels of *LsLFY* and *LsAP1* [59], which expression is most abundant at the onset of bolting [58,60]. Induction of high *LsFT* expression during the transition to reproductive growth and activation of *LsLFY* and *LsAP1* was also observed in three heading and non-heading lettuce varieties grown in the field in natural conditions [60].

Transcriptomic data from lines that are either bolting resistant or sensitive to high temperature, identified floral integrator genes like *LsSOC1*, *LsFT* and *LsAP1* as upregulated in the bolting sensitive line. [61]. Gene expression analysis of shoot apical meristem cells undergoing flowering transition in response to high temperature on the bolting-sensitive lettuce line S39, and further gene function studies, confirmed a role of *LsSOC1* in heat-promoted bolting in lettuce [36]. When expressed from the 35S promoter, *LsSOC1* acts as an activator of flowering in *A. thaliana* and can fully complement the Arabidopsis *soc1* null mutant [36]. RNAi-mediated knockdown of *LsSOC1* in *L. sativa* results in late flowering plants with reduced *LsLFY* expression [36]. The important function of *LsSOC1* in heat induced bolting in lettuce was further supported by the identification of two heat shock transcription factors that bind to the promoter of *LsSOC1* [36].

Overall, *LsFT* and *LsSOC1,* and their putative floral meristem identity targets *LsAP1* and *LsLFY* seem to play a key role in flowering transition in *L. sativa*, similar to other plant species. However, the key role of *LsSOC1* in promoting heat-induced flowering was not observed in other species so far, and may constitute a unique feature whose conservation amongst other Asteraceae species should be investigated.

### 4.2. Overview of the Vernalization and Autonomous Pathways

Vernalization refers to a process by which prolonged period of cold (winter) renders plants competent to flower, often many weeks later when other conditions, like day length or ambient temperature, are favorable [62]. Duration of cold exposure and the optimal temperature for vernalization vary among species, and among ecotypes of a given species, as plants adapt to periods of cold that are typical of a winter season in their natural habitat [8]. Plants can be either annual, biennial or perennial depending on the time required to complete their life cycles, from germination to seed setting, and the length of vegetative phase. Perennial plants can reproduce several times with recurrent vegetative to flowering cycles, and often do not respond to vernalization in the first year(s) of life. In annuals and biennials, vegetative to reproductive transition occurs once and flowering is associated with senescence and death of the whole plant [63].

In Arabidopsis, two genes are responsible for much of the variation in flowering time among natural population, *FLOWERING LOCUS C* (*FLC*), which acts as a repressor of flowering, and *FRIGIDA* (*FRI*), which promotes expression of *FLC*. In response to prolonged exposure to low temperatures, *FLC* is progressively repressed through epigenetic and silencing mechanisms, leading to flowering response. *VERNALIZATION INSENSITIVE3* (*VIN3*), a factor needed for epigenetic silencing of *FLC*, was recently found to have a key and complex role in vernalization and response to different temperatures [64]. These studies indicate that the absence of warmth rather than the presence of cold might be necessary for vernalization. Pivotal roles of *FLC* and *VIN3* in flowering time adaptation to natural environments were also confirmed by genome-wide association studies with nearly complete genotype information from 1135 Arabidopsis accessions [65].

The vernalization response is largely conserved within the Brassicaceae species due to conserved function of the main regulators FLC and FRI. However, the complex rearrangements occurred in the Brassica genomes [66,67,68] likely led to neofunctionalization processes of some *FLC* and *FRI* paralogs, which have lost their role in flowering control in response to vernalization.

In perennial Brassicaceae (e.g., *Arabis alpine*), orthologs of *FLC* are repressed by winter cold and reactivated in spring conferring seasonal flowering patterns, differently from annuals where they are stably repressed by cold as in Arabidopsis. Sequence comparisons of *FLC* orthologs from annuals and perennials identified two regulatory regions in the first intron whose sequence variation correlates with divergence of the annual and perennial expression patterns [69]. Unstable repression of a *C. intybus FLC* homolog during the cold season was also confirmed in root chicory that is perennial [22]. This points to key role of *FLC* regulation in evolutionary transitions between perenniality and annuality that seems to have occurred often among higher plants.

Questions regarding flowering response to vernalization in *Lactuca* species remain open as cultivated plants seem to have lost the need for the vernalization that is present in wild relatives. More generally, several species in the Asteraceae family require vernalization to flower, however molecular mechanisms underlying this trait have been poorly investigated.

Overall, null mutations or decreased expression of either *FLC* or *FRI,* as well increased expression of *FLC* negative regulators, would result in early and vernalization-independent flowering induction.

#### 4.2.1. Vernalization and Autonomous Pathway in Brassicaceae

In *A. thaliana,* winter annuals contain active alleles at two loci, *FLOWERING LOCUS C* (*FLC*) and *FRIGIDA* (*FRI*), whereas summer annuals harbor inactivating mutations in one or both of these genes [70,71,72,73]. FLC is a MADS box transcription factor that acts as a repressor of flowering by directly binding to the floral promoting genes *FT*, *SOC1* and *SQUAMOSA PROMOTER-BINDNG PROTEIN-LIKE* 15 (*SPL15*) to block their transcription [40,70] (Figure 4). *FRIGIDA* encodes a coiled-coil protein that promotes *FLC* transcription, probably by affecting its chromatin structure [72]. During cold treatment, *FLC* is repressed through chromatin remodeling [74], and epigenetic mechanisms maintain the repressed state of *FLC* upon return to higher temperatures. [75]. During vernalization, transcription of several long noncoding RNAs (lncRNAs) starts from sites within the intron (COLDAIR) and promoter of FLC (COLDWRAP) and a set of antisense transcripts of *FLC,* collectively named COOLAIR, are induced and physically associate with the *FLC* locus. This accelerates the transcriptional shutdown of *FLC* by recruitment of chromatin remodelers and switching of chromatin states [76,77,78,79]. Histone modifications mediated by genes like *VERNALIZATION 1* (*VRN1*), *VRN2, VERNALIZATION INSENSITIVE 3* (*VIN3*), cooperate to repress *FLC* at chromatin level [80,81,82,83,84]. FLOWERING LOCUS CA (FCA), FLOWERING LOCUS D (FLD), FLOWERING LOCUS KH DOMAIN (FLK), FLOWERING LOCUS PA (FPA), FLOWERING LOCUS VE (FVE), FLOWERING LOCUS Y (FY), and LUMINIDEPENDENS (LD) also repress *FLC* to accelerate flowering independently of vernalization. The corresponding genes are part of the so-called autonomous flowering pathway and act through repressive chromatin remodeling complexes and small RNAs to negatively regulate *FLC* [8]. FLC-like proteins form a specific phylogenetic clade, some members of which (MADS AFFECTING FLOWERING, MAF) can form protein complexes with FLC and redundantly affect flowering in response to vernalization [85].

Many Brassica species are biennial and require vernalization at seedling or mature plant stage. The temperature and duration of vernalization varies between spring, semi-winter and winter type plants: Flowering occurs without vernalization in spring types, with low vernalization (exposure to cold for shorter periods) in semi-winter types and with longer exposure to cold temperature in winter types. Rapid cycling populations, with extremely short reproductive cycles and which flower early independent of vernalization, have been developed in different Brassica species [86]. Comparative phylogenetic analysis of *B. napus*, *B. rapa* and *B. oleracea* identified three FLC clades which reflects the whole-genome triplication events that occurred during the evolution of the Brassica genome [66,87]. Four *FLC* paralogs in *B. rapa* (*BrFLC*) [67], five in *B. olearacea* (*BoFLC*) [68] and nine in *B. napus* (*BnaFLC*) [66] were identified (Table 3). *FLC* homologs in the chromosome A10 and C02 of *B. napus*, and an additional one in A03 (*Bna.FLC.A03b*), were initially associated with flowering time in *B. napus*. However, genome-wide association studies of flowering time and vernalization response in 188 different accessions demonstrated that *Bna.FLC.A02* and *Bna.FLC.C02* account for a significant proportion (22%) of natural variation in diverse accessions [50]. Expression of eight out of nine *BnaFLC* genes were downregulated during vernalization. This suggests that vernalization modulates *FLC* expression levels in a similar manner as in Arabidopsis. A cold-responsive *FLC-FRI-CBF1* cluster including *Bna.FLC.A03b* and *Bna.FRI.A3/Bna.FRI.Xa* was identified. It has been shown in other species that gene clusters with functionally related genes might be maintained by selection pressure to enable adaptation to extremely diverse environments in a similar manner as the cold-responsive cluster *FLC-FRI-CBF1* [88,89]. *Bna.FLC.A03b* shows enhanced expression levels in winter compared to semi-winter type plants [66]. Four *FRI* possible orthologs were identified in *B. napus.* [14]. Association analysis in a double-haploid population revealed that six SNPs (Single Nucleotide Polymorphism) in *Bna.FRI.A03* are associated with flowering time variation in 248 accessions, and that specific haplotypes are over-represented in semi-winter or winter types, while spring type plants did not show this correlation [3,14,90]. These data suggest that *Bna.FLC.A03b* and *Bna.FRI.A03* are functionally related, similar to *FLC* and *FRI* in *A. thaliana*, and have a key role in *B. napus* flowering response to vernalization.

In *B. rapa*, both *Bra.FLC.A10* (*BrFLC1*) and *Bra.FLC.A02* (*BrFLC2*), were found to underlie QTLs for flowering time in different studies [49,91,92,93,94], possibly due to alternative splicing and a 57 InDel (INsertion/DELetion) leading to a non-functional allele [91], respectively. However, the most similar *B. rapa* homolog of *Bna.FLC.A03b*, *BrFLC5*, is truncated and is expected to be not functional [95]. A recent study showed that the reference genome sequence indeed contains a truncated *BrFLC5* sequence, while other accessions contain functional genes with different splicing patterns resulting from a single nucleotide mutation. Genetic variation within the *BrFLC5* locus indicates that *BrFLC5* is not a major regulator of flowering time [96]. *BrFLC2* acts as a repressor of flowering when overexpressed in *A. thaliana* and shows early flowering when silenced in *B. rapa ssp. chinensis* (Pak-choi) [95,97]. *BrFLC2* seems to negatively regulate flowering by enhancing *MADS AFFECTING FLOWERING2* (*BrMAF2*) expression, while inhibiting expression of *BrSOC1* and *BrSPL15* [97]. In *B. rapa* seedlings, *BrFLC2* expression levels decrease upon vernalization treatment and remained low after return to higher temperatures. Contrarily to *BrFLC2*, expression of *BrVIN3*, a negative regulator of *FLC*, is very low in 14-day-old seedlings without vernalization, activates after four-week vernalization treatment on seeds and decreases again after transfer to higher temperature [95].

At chromatin level, *BrFLC* genes contained active chromatin marks H3K4me3 and H3K37me3 under normal growth conditions. During vernalization, alternative splicing of five *BrCOOLAIR* transcripts (*BrFLC2as406*, *-477*, *-599*, *-755* and *-816*) reduced H3K37me3 levels of *BrFLC1*, *BrFLC2* and *BrFLC3*. Differently from the Arabidopsis *COOLAIR*, *BrCOOLAIR* is located further downstream of *BrFLC2* and, during vernalization, class II transcripts, which are polyadenylated in the region complementary to the *BrFLC* promoter, are more abundant than class I, which are polyadenylated in the region complementary to the last intron of *BrFLC* [98]. Together with reduced H3K37me3 levels, an increase of H3K27me3 was detected in *BrFLC1*, *BrFLC2* and *BrFLC3* upon vernalization, which was maintained when plants were transferred to higher temperatures [95]. Besides affecting *FLC*, vernalization also resulted in enhanced H3K27 methylation in *BrMAF1* and DNA demethylation of two subunits of *casein kinase II* (*CK2*), *BrCKA2* and *BrCKB4*, altering daily expression period of clock-related gene *CIRCADIAN CLOCK-ASSOCIATED1* (*BrCCA1*) [95,99]. These findings indicate that the mechanisms underlying vernalization in *B. rapa* are very similar to those of Arabidopsis and involve chromatin modifications and COOLAIR antisense transcription.

In *B. oleracea*, expression levels of *BoFLC2* and *BoVRN* are enhanced in early compared to late flowering *B. oleracea* genotypes when grown at ambient temperature (22.5 °C and 12/12 h light/dark period) [100]. Two alleles for *BoFLC4* are described, which both confer a requirement for vernalization but respond with different kinetics to temperature shifts. Plants containing allele E9 require longer cold periods and flower late compared to those harboring allele E5. Introduction of genomic fragments containing the *BoFLC4^E5^* or *BoFLC^E9^* allele complemented an Arabidopsis *flc* null mutant, with a stronger effect for *BoFLC^E^*^9^ [101]. The closest *B. oleracea* ortholog of *BnaXFRId*, *BoFRIa*, also acts as a repressor of flowering when transformed into an Arabidopsis *fri* null mutant [90,101,102]. This indicates that FLC and FRI function in vernalization is also conserved in *B. olearacea*.

*R. sativus* is not a vernalization-requiring plant, but cold treatment does accelerate flowering. Radish transcriptome analysis during vernalization resulted in the identification of several vernalization-related differentially expressed genes [18]. Three copies of *RsFLC* were detected and all three act as flowering repressors when overexpressed in *A. thaliana* [103]. *RsFLC* expression before vernalization was enhanced in a late- compared to an early-bolting *R. sativus* inbred line, and reduced during vernalization or after GA treatment [18,104,105]. Overall, negative regulators of the vernalization pathway, such as *RsFLC, RsMAF2, RsSPA1*, and *RsAGL18*, were highly expressed in the late-bolting line, whereas positive regulators of vernalization, such as *RsVRN1*, *RsVIN3*, and *RsAGL19* were relatively highly expressed in the early-bolting line [104]. These results suggest that the vernalization pathway is conserved between radish and Arabidopsis.

*D. tenuifolia* is not a vernalization-requiring plant, and cold treatment of either seeds or plantlets does not accelerate flowering. Hence, even though *DtFLC* acts as repressor of flowering when overexpressed in *A. thaliana* and can complement the Arabidopsis *flc* null mutant, its role as a regulator of flowering time in wild rocket has to be further investigated [19,106].

#### 4.2.2. Vernalization and Autonomous Pathway in Asteraceae

Wild lettuce-related species like *L. virosa* require vernalization to induce flowering. The cultivated *L. sativa* does not require a cold treatment for flowering, but a few days of cold does result in a better germination. Expression of the lettuce homolog of *FVE*, *FLD* and *LD* of the autonomous pathways were found to correlate with *LsFT* expression and flowering induction in two early or late *L. sativa* varieties grown in the field [60]. This finding suggests the existence and function of the autonomous pathway in lettuce flowering induction. However, the expression of lettuce *FLC* homologous genes was not analyzed either in this or in other studies, which impedes any further consideration about a possible role of *FLC*-like genes in cultivated lettuce.

A *FLC*-like gene, *CiFL1*, was identified and studied in *C. intybus*, which is biennial and requires vernalization at seedling or mature plant stage. Overexpression of *CiFL1* in Arabidopsis causes late flowering and prevents upregulation of the *AtFLC* target *FLOWERING LOCUS T* by photoperiod, suggesting functional conservation between root chicory and Arabidopsis [107]. *CiFL1* was repressed during vernalization of seeds or plantlets of chicory, like *AtFLC* in Arabidopsis. However, *CiFL1* repression was not maintained when plants were returned to warmer temperatures. This may be linked to the perenniality of root chicory compared with the annual life cycle of Arabidopsis. [22]. Indeed, recent studies on the divergence of seasonal flowering behavior among annual and perennial species in Brassicaceae showed that in perennial Brassicaceae orthologs of *FLC* are repressed by winter cold and reactivated in spring conferring seasonal flowering patterns, whereas in annuals, they are stably repressed by cold [69].

### 4.3. Overview of the Ambient Temperature Pathways

Responsiveness to ambient temperature is an adaptive trait and varies widely between and within species and accessions [108]. Besides extreme changes in temperature (e.g., vernalization), small changes in ambient temperature can also have an effect on flowering time. In *A. thaliana* plants grown under controlled laboratory conditions, a shift to lower (23 °C to 16 °C) and higher (23 °C to 27 °C) temperature delays and enhances flowering time, respectively [109]. The MADS box transcription factor *SHORT VEGETATIVE PHASE* (*SVP*) and most genes from the *FLC* clade, such as *FLOWERING LOCUS M* (*FLM*/*MAF1*) and *MADS AFFECTING FLOWERING-2-4* (*MAF2–MAF4*), have been implicated in the thermosensory pathway [85,107,110], with *SVP* and *FLM* having key roles in this process in Arabidopsis (Figure 4). SVP represses *FT* transcription at lower temperatures, but the levels of *FT* mRNA increase at higher temperatures. The control of floral transition in response to ambient temperature seems to differ among plant species, and many important questions concerning the regulation of flowering time by ambient temperature in Arabidopsis remain unsolved. However, *FT*-like genes seem to integrate the response to changes in ambient temperature in many species [111].

Although flowering induction in response to temperature changes may greatly affect yield and product quality in both Brassicaceae and Asteraceae crop species, insufficient work has been done to identify the genes responsible for this trait, especially in Brassica species. The floral integrator SOC1 was suggested to mediate heat-promoted bolting in lettuce, but further studies are needed to establish the exact mechanisms of flowering induction under these conditions, and whether this role is conserved in other Asteraceae [36].

Mutations that increase or decrease the expression of *MAF*s and *SVP* genes, known to be negative regulators of flowering time in Arabidopsis, may delay or speed up flowering time, respectively, if molecular mechanisms were conserved in crop species. On the other hand, reduction of *SOC1* in Asteraceae would potentially result in delayed timing of bolting and insensitivity to high temperature.

#### 4.3.1. Ambient Temperature Pathways in Brassicaceae

Ambient temperature affects the deposition of the histone variant H2A.Z by the chromatin remodeling factor ACTIN RELATED PROTEIN 6 (ARP6). H2A.Z has been proposed to compact DNA in a temperature-dependent manner, thereby functioning as a temperature sensor in *A. thaliana* [112]. Accordingly, *arp6* mutants display a constitutive warm temperature response, but are still temperature responsive, indicating that H2A.Z is not the only thermosensor that mediates flowering. Recently, the basic helix-loop-helix (bHLH) transcription factor PHYTOCHROME INTERACTING PROTEIN 4 (PIF4) was shown to mediate flowering in response to temperature downstream of H2A.Z [113]. Mutations in *PIF4* suppress the induction of flowering by high ambient temperature only in SD, whereas the *pif4* mutant flowers normally in inductive LD [114]. The response to 27 °C-SD in the leaves was found to depend on the coordinate functions of CO, PIF4 and PIF5, as well as SVP, providing a genetic and molecular framework for the interaction between the photoperiod and thermosensory pathways [115].

*SVP* is directly activated by the chromatin remodeler BRAHMA (BRM) during the vegetative phase, whereas *FLM* is also regulated by the vernalization and photoperiodic pathways (reviewed in [116]). SVP can interact with FLC or FLM to form a repressor complex to prevent the expression of *FT* and *SOC1* [117,118]. Loss-of-function of *SVP* or *FLM* results in early and temperature-insensitive flowering, although *flm* loss-of-function plants retain some temperature sensitivity below 10 °C [118]. *FLM* is subject to temperature-dependent alternative splicing [110]. Two most abundant splice forms of *FLM*, *FLMβ* and *FLMδ*, which differ in the incorporation of either the second or third cassette exon, are both translated into proteins and their splicing pattern changes in response to changes in ambient temperature [110,118,119,120,121]. Different studies have shown that the abundance of *FLM-β* and *FLM-δ* splicing variants is regulated by temperature in an opposite fashion, with *FLM-β* enhanced at low temperature (16 °C) and *FLM-δ* increased at high temperature (27 °C) [118,120]. Overexpression of either *FLM-β* or *FLM-δ* results in opposite phenotypes, with *FLM-β* overexpression delaying flowering, as expected for a floral repressor, and overexpression of *FLM-δ* accelerating the transition to flowering [116,120]. A model was proposed in which only the incorporation of the FLM-β protein in the SVP–FLM complex would result in active repression of flowering targets, whereas incorporation of *FLM-δ* would form an inactive complex, indirectly promoting the transition to flowering [120]. More recent studies have shown that splice variant *FLM-β* has a stronger effect on flowering time compared to *FLM-δ* and therefore the function of *FLM-δ* under natural conditions is a matter of debate [108,122]. *SVP* and *FLM* contribute to the variation of flowering time among natural accessions of *A. thaliana* [73,123]. Alternative splicing is an important mechanism in sensing and adapting to changes in ambient temperature, and several genes in the thermosensory pathway undergo alternative splicing in response to temperature changes [121]. *MAF2*, *MAF3*, and circadian clock associated genes *PRR7* and *CCA1*, showed alternative splicing variants after a temperature shift.

Genes homologous to *SVP* and *FLM/MAF1* have been identified in *B. napus*, *B. rapa*, *B. oleracea* and *R. sativus*. In *B. rapa*, *BrSVP* and *BcMAF1*, a MAF-related Pak-choi (*B. rapa ssp chinensis*) gene, cause late flowering when transformed individually into *A. thaliana* [3,107]. Silencing of *BcMAF1* in Pak-choi resulted in enhanced expression of *BcFT1*, *BcFT2* and *BcSOC1*, reduced expression of *BcMAF2* and early flowering compared to control plants [3]. These findings point to a function of SVP and FLM/MAF1 in the regulation of flowering time, but their role in ambient temperature response was not explored.

*R. sativus* plants flower early in spring, with LD conditions and higher temperature, compared to autumn. Vernalization and LD conditions reduces *RsSVP* expression, while expression is enhanced in SD conditions [105], indicating that RsSVP may act as a repressors of flowering in radish, as in Arabidopsis.

In *B. oleracea*, shifting plants to higher (23 °C to 27 °C) temperature results in differential splicing of about 156 genes. However, only 1% to 2.2% of those overlap with transcripts that are differentially expressed in the two investigated *A. thaliana* accessions (Gy-0 and Col-0). In contrast to *A. thaliana*, no differential splicing in flowering time genes was described in *B. oleracea* in response to high temperature [121], indicating that alternative splicing may not be a general regulatory mechanism by which ambient temperature regulates flowering response in Brassica species other than Arabidopsis.

#### 4.3.2. Ambient Temperature Pathways in Asteraceae

*L. sativa* plants grown at high temperatures (35/25 °C) flower early compared to plants grown at lower temperatures (25/15 °C) [58]. RNA-seq analysis revealed 1443 and 1216 genes that were upregulated respectively in leaves and stems of plants that had been shifted to 37 °C for one week compared to control plants that were maintained at 25 °C [124]. Among these genes were homologs of *AP2*, *AP2*-like, *SOC1* and *FLM* in the leaves and homologs of *AP2*-like, *FLC* and *FLM* in the stem. The shift to 37 °C resulted in the downregulation of 1038 genes in leaves and of 933 genes in stems, as compared to the controls at 25 °C. These included photoperiod-related genes in both leaf and stem, and two *LsFLC*-like homologs in leaf. Unexpectedly, *SVP*-like genes were not present in the sets of differentially regulated transcripts [124].

In *C. intybus*, treatment of non-vernalized plants with elevated temperatures (increase of 6 °C) in the field resulted in a variety of phenotypic differences like more leaves, reduced mean leaf area, decreased root weight and early flowering [125]. The severity of these heat stress-induced phenotypic changes was cultivar dependent. Early flowering in response to elevated temperature seems to be conserved in *L. sativa* and *C. intybus*. However, no genetic or molecular data are available in *Cichorium* spp. for heat-induced bolting response.

### 4.4. Overview of the Photoperiodic Pathway

Day length is an important factor for a plant to track seasonal changes, where short days (SD, 8/16 h light and dark) indicate winter and long days (LD, 16/8 h light and dark) indicate spring or summer. Plants can be divided into three major groups on the basis of their responses to photoperiod: Long-day plants flower when the day exceeds a critical length, short-day plants flower when the day is shorter than a critical length and day-neutral plants flower independently of day length [126]. As plants aim to flower in the optimal season, most plants show a delay in bolting when grown under SD conditions and early bolting under LD conditions. The mechanism behind light perception and integration has been intensively studied in *A. thaliana* over the past 15 years (reviewed in [11] and [127]). The circadian clock and photoreceptors influences transcription and protein stability of the transcriptional activator *CONSTANS* (*CO*) which, in a signaling cascade involving GIGANTEA (GI), in turn activates the floral integrator *FT* in a long-day afternoon [39,128].

Photoperiod and circadian rhythm are involved in many processes of adaptive response to environmental conditions, including flowering time. Their molecular mechanisms are widely conserved amongst plant species to such an extent that mechanisms of photoperiod measurement are more diverse between long-day and short-day plants than between eudicots and monocots [129]. Based on gene expression, it is suggested that the photoperiod pathway is conserved between the Brassicaceae and Asteraceae family, which include mainly plants requiring long days to flower. Despite our knowledge on the genetic control of flowering time in response to different light conditions is quite limited in the species we are reviewing, preliminary studies suggest a key role of CO, GI and photoreceptors in adaptation to different environments [54,106,130].

#### 4.4.1. The Photoperiodic Pathway in Brassicaceae

CONSTANS promotes flowering by initiating transcription of the *FT* and *TSF* genes (Figure 4). The blue light receptor FLAVIN-BINDING KELCH REPEAT F-BOX 1 (FKF1) and the clock-associated protein GI form a complex to degrade transcriptional repressors of *CO*, *CYCLING DOF FACTORs* (*CDFs*), and to stabilize the CO protein [131,132,133,134,135]. Post-translational regulation of CO is essential for a flowering response to long days. The CO protein is ubiquitylated by a ubiquitin ligase complex that includes CONSTITUTIVE PHOTOMORPHOGENIC 1 (COP1) and SUPPRESSOR OF PHYTOCHROME A (SPA1), facilitating CO degradation by the 26S proteasome [136,137,138]. Activity of this complex is repressed by light so that it mainly promotes the degradation of CO protein in the dark. Thus, only the peak of *CO* mRNA that occurs in the light at the end of a long day after degradation of the CDFs by GI–FKF1 leads to CO protein accumulation (Figure 4).

The circadian clock is a time-keeping mechanism with a periodicity of 24 h. In Arabidopsis, the circadian clock confers diurnal patterns of gene expression on roughly one-third of the genes, and comprises interlocked feed-back loops [139,140]. Core clock components include the morning phased genes *CCA1*, *LATE ELONGATED HYPOCOTYL* (*LHY*), *REVEILLE8* (*RVE8*) and *PSEUDO-RESPONSE REGULATOR 9* (*PRR9*) [141,142,143,144,145,146,147]; the afternoon phased genes *PRR5*, *PRR7*, *GI* [145,147]; and the evening phased genes *EARLY FLOWERING 3* (*ELF3*), *ELF4*, *LUX ARRHYTHMO* (*LUX*) and *TIMING OF CAB EXPRESSION1* (*TOC1*) [142,147,148,149,150].

Homologs of all genes involved in photoperiodic response were identified in *B. napus* and shown to be highly variable in studies of targeted deep-sequencing of essential flowering time regulators [35] in a panel of 280 inbred lines. Four *CO* and four *CO*-like genes are present in the genome of *B. napus*, including those initially characterized by Robert et al. [151], one of which shown to complement *co* mutants in *A. thaliana* [151]. One *BnPHYA* gene has undergone two coupled duplication-deletion events (HNRTs), where one region of the genome replaces a respective homeologous genome region. It was suggested that such rearrangements may represent a necessary co-adaptation of the photoperiodic pathway to the strong vernalization requirement in winter inbred lines [4].

Compared to *A. thaliana*, several duplicated or triplicated photoperiod genes, such as *BrCO*, *BrFKF1*, *BrCDF1*, *BrLHY* and *BrTOC1*, were detected in *B. rapa* [35,54]. Expression of these genes throughout the day differed when plants were grown under LD or SD conditions, only *BrCDF1* showed a similar trend under both growth conditions [54]. So far, no complementation or other functional studies for the core clock components are available in Brassica species. *BrGI* was identified as an important component for circadian rhythm and multiple abiotic stress responses and acts as an activator of flowering when transformed into an Arabidopsis *gi* null mutant [152]. Two putative null alleles of *BrGI* resulted in late flowering when homozygous in *B. rapa* [152,153]. Furthermore, BrGI protein physically interacts with GI-interacting partners, like BrFKF1, suggesting a conserved function with Arabidopsis [152].

In both *B. oleracea* and *R. sativus*, silencing of *GI* resulted in delayed bolting and flowering, with a correlation between *GI* expression levels and days to flowering [154,155].

*D. tenuifolia* plants flower later under SD compared to LD conditions, with 50 and 20 days to flowering, respectively [106]. *DtCO* and *DtGI* are both diurnally regulated. Under LD conditions, *DtCO* acts as activator of flowering when transformed into *A. thaliana* and could complement the *co* null mutant [106].

#### 4.4.2. The Photoperiodic Pathway in Asteraceae

Lou et al. [156] hypothesized that *CCA1*, *RVE2*, *RVE4* and *RVE5* function might be restricted to the Brassicaceae family. However, Higashi et al. [130] later described 215 common oscillating transcripts in *L. sativa*, including *LsCCA1*, *LsGI*, *LsLHY*, *LsFKF1*, *LsTOC1*, *LsPRR7* and *LsCO-*like. The expression pattern of these genes show a large degree of overlap with those of *A. thaliana* [130], indicating a possible functional conservation in Asteraceae. Despite the great importance of photoperiodic control of flowering time for vegetable crop production and adaptation to different cultivation environments, no further molecular and genetic data are available for either lettuce or chicory species.

### 4.5. Overview of the Age Pathway

Plants go through developmental phases such as juvenile-to-adult transition and floral induction during their life cycle. As the plant ages, concentrations of the SQUAMOSA PROMOTER BINDING LIKE (SPL) transcription factors (also known as SQUAMOSA promoter binding protein, box family, SBP) increase. SPLs promote flowering by initiating the expression of several other transcription factors, such as LEAFY (LFY), FRUITFULL (FUL), and SOC1 [157,158]. SPL proteins are negatively regulated by the microRNAs [158]. MicroRNAs (miRNA) are key regulators of the age pathway, preventing precocious flowering when the plant is too young. Two major miRNAs, *miR156* and *miR172*, have an antagonistic effect on flowering time by downregulating their own set of target genes. *miR156* expression is high in young plant stage, decreases over time and is low at the onset of flowering [159,160].

The involvement of miRNAs in flowering time and the important role of *miR156* and *miR172* and their corresponding targets, is widely conserved across plant species [161]. As expected, both *miR156* and *miR172* seem to be conserved between the Brassicaceae and Asteraceae families, although very few reports are available in Asteraceae. The miR156/SPL module plays a central function in age-dependent competence to flowering, but seems to be even more fundamental in perennial Brassica species that undergo reiterative flowering induction cycles and do not respond to vernalization in the first year of life. Therefore, miR156/SPL may play a key role in flowering control in biennial crops [162]. Other miRNA like *miR824* and *miR5227,* the latter only detected in *R. sativus* [163], are less conserved and seem to be newly evolved Brassica-specific miRNAs as they were not found in families other than Brassicaceae so far. In *L. sativa*, a homolog of the Arabidopsis DELAY OF GERMINATION1 (DOG1) seems to have acquired a novel function in the miRNA-mediated response to flowering time, but further studies are needed to investigate DOG1 role in other Asteraceae and in other plant families [164].

#### 4.5.1. Age Pathway in Brassicaceae

*A. thaliana* contains eight *miR156* members (*miR156a* to *miR156h*) which target different *SPL* genes (Figure 4) [159,165]. Besides enhancing expression of floral meristem identify genes, *SPL* genes also promote *miR172* expression [166]. *miR172* shows an inverse expression pattern with increasing expression over time [159,160]. *A. thaliana* contains five *miR172* members (*miR172a* to *miR172e*) which target *AP2* and the *AP2-*like genes *TARGET OF EAT1* (*TOE1*), *TOE2*, *TOE3*, *SCHLAFMÜTZE* (*SMZ*) and *SCHNARCHZAPFEN* (*SNZ*) [167,168,169]. *AP2* and *AP2*-like genes inhibit the onset of flowering by repressing expression of *SOC1*, *FUL* and *AGAMOUS* (AG) (Figure 4, [170]). Another miRNA, *miR824*, targets *AGL16*, which encodes a MADS-box repressor of flowering time that interacts with SVP and FLC to regulate *FT* expression levels [171]. SPL15 cooperates with SOC1 to coordinate the basal floral promotion pathways required for flowering in non-inductive environments by directly activating transcription of *FUL* and *miR172* in the SAM [162]. The capacity of SPL15 to promote flowering is regulated by age through *miR156* that targets *SPL15* mRNA. Strong evidence is emerging that miR156/SPL control competence to flower as well as vegetative phase change [162]. Several studies point to a major role of SPL9 and SPL15, with SPL15 playing the larger role in floral induction, particularly under noninductive short days, and SPL9 acting in floral primordia after the floral induction. The miR156/SPL module is of special interest for the acquisition of competence to flowering in biennial and perennial Brassicaceae relatives of Arabidopsis, where *miR156* levels act as the timer in controlling competence to flower, and often make plants insensitive to vernalization when too young (Figure 4, [172,173]). It was suggested that the miR156/SPL module, which is evolutionarily conserved in all flowering plants, might have acquired increased dependency for flowering in perennials, whereas annuals would have evolved genetic mechanisms to bypass this module by alternative inductive pathways such as light/photoperiod [162].

*B. napus* contains 36 copies of *miR156*, of which 17 located on the A genome and 19 on the C genome, and 14 copies of *miR172*, with eight located on the A genome and six on the C genome [174]. A total of 58 genes encoding putative SPL/SBP proteins are present in the *B. napus* genome, 44 of which harboring *miR156* binding sites [175]. This suggests that relationship between *miR156* and *SBP* genes is conserved across species, although distinct regulation pattern of the homologous genes exist between *B. napus* and Arabidopsis that may reveal some divergence of the SBP-box genes in oilseed rape.

*B. rapa* contains 17 copies of *miR156* and 11 of *miR172* [174]. *BrmiR156* is highly expressed in early plant stages and expression decreases during plant development. *BrSPL9-2* and *SPL15-1* show an opposite expression pattern compared to *BrmiR156*, with increasing expression over time. Cabbage plants expressing a mutated *BrSPL9* (*mBrSPL9*) allele, resistant to *Brm*iR156, showed enhanced *BrSPL9* and *BrmiR172* expression. In the field, *mBrSPL9* plants had dark green leaves with enhanced chlorophyll content and a prolonged heading stage with delayed flowering, but no significant change in head weight, size or shape. Overexpression of *BrmiR156* in cabbage resulted in decreased *BrSPL9*-2 transcript levels and a prolonged seedling and rosette stage [176], pointing to conservation of the miR156/SPL module in *B. rapa*.

*B. oleracea* contains 15 copies of *miR156*, where *BomiR156c* is known to target *BoSPL9* while *BomiR156g* targets *BoSPL3*. The *miR172* family contains nine copies and targets *BoAP2* and *BoTOE2* [174,177,178]. A newly evolved *miR824*, which seems specific for Brassicaceae, was also identified and targets *BoAGL16* [178]. This function is conserved with Arabidopsis where the miR824/AGL16 quantitatively modulate the extent of flowering time repression in a long-day photoperiod through *FT* [171].

In *R. sativus*, 11 members of *miR156*/*miR157*, five members of *miR172*, two members of *miR824* and one member of *miR5227* are detected. Different *RsmiR156* copies target *RsSPLs* and *RsmiR156a* also *RsTOC1, RsmiR172a* targets *RsAP2, RsmiR824* targets *RsAGL16* and *RsmiR5227* targets *RsVRN1*. Expression of *RsmiR156a*, *RsmiR824* and *RsmiR5227* decreased when plants shifted from vegetative to reproductive phase [163], strongly indicating that these miRNAs and their corresponding target genes might play important roles during bolting and flowering processes of radish.

#### 4.5.2. Age Pathway in Asteraceae

*LsmiR156* and *LsmiR172* act as repressor and activator of flowering, respectively, when expressed in *A. thaliana* [164], and targets *LsSPLs* and *LsAP2* in *L. sativa* [179]. In Arabidopsis, expression of the *DELAY OF GERMINATION1* (*DOG1*) gene responds to seed maturation temperature and determines the depth of seed dormancy [180]. Huo et al. showed that DOG1 could regulate seed dormancy and flowering times in lettuce through the modulation of *miR156* and *miR172* levels [164]. *LsDOG1* silencing lines flowered early compared to control *L. sativa* plants, with an enhanced effect in autumn, and showed reduced expression of *LsmiR156*, enhanced expression of *LsmiR172*, *LsFT*, *LsSPL3* and *LsSPL4* and no difference in transcript levels of *LsSPL9*. This would suggest that *LsDOG1* has an additive role in *LsmiR156*- and *LsmiR172*-mediated flowering time, besides the thermo-inhibition of seed germination described in *A. thaliana* [164].

Srivastava et al. [181] has predicted two copies for *miR156* and one copy for *miR157* in *C. intybus*. For the miRNA targets, only *CiSPL3* and *CiSPL12* were detected and confirmed as targets of *CimiR156* [181].

### 4.6. Overview of the Hormonal Pathway

Gibberellins (GAs) are growth regulators involved in plant developmental processes that promote transition to flowering in several plant species [61,124,182,183,184,185]. In Arabidopsis, GA contributes to flowering under inductive long days (LDs) through the activation of *SOC1* and *LFY* in the inflorescence and floral meristems, and of *FT* in leaves. Under non-inductive short days (SDs) conditions, the GA pathway assumes a major role as under SDs flowering is delayed and correlates with a gradual increase in bioactive GA at the shoot apex [186]. Mutations that impair GA biosynthesis prevent flowering under SDs [183].

Besides GA, it has been suggested that cytokinins, major growth regulators in plants, also participate in the regulation of flowering time (reviewed in [187]). For a long time, it has been known that exogenously applied cytokinin can promote flowering in Arabidopsis [188,189,190]. However, it is unclear whether endogenous cytokinins can also have the same inductive activity.

Regulation of GA and its involvement in the switch to flowering seems conserved between the Brassicaceae and Asteraceae families. In grasses and cereals, GAs are similarly regulated and also involved in flowering time [191], suggesting that the GA role in promoting flowering is widely conserved in plants. Genes involved in GA metabolism or sensitivity may constitute good targets to modulate flowering time in crops, as enhanced GA content or signaling can induce early flowering whereas low GA amount or signal can delay bolting and flowering.

#### 4.6.1. Hormonal Pathway in Brassicaceae

The GA pathway is well described in *A. thaliana* (reviewed in [192]). In brief, the last steps of the GA pathway involves the conversion of GA_12_ into GA_9_ and GA_53_ into GA_20_, by GA 20-oxidases (GA20ox1-5), the conversion of GA_9_ and GA_20_ into bioactive GA_4_ and GA_1_, respectively, by GA 3-oxidases (GA3ox1-3) and the deactivation of GA_4_ and GA_1_ by GA 2-oxidases (GA2ox1-5) [193,194,195,196,197,198]. Bioactive GAs binds to *GIBBERELLIN INSENSITIVE DWARF1* (*GID1a*, -*b* and -*c*) to promote degradation of DELLA proteins [199,200,201], negative regulators of gibberellin signaling that act immediately downstream of the GA receptor. DELLA proteins repress transcription of many genes, including *FT*, *TSF*, and some *SPL* genes [202]. Low levels of bioactive GA result in the accumulation of DELLA proteins, which delay flowering independent of photoperiod [202,203]. The MADS box transcription factor SVP, besides repressing floral integrator gene expression, regulates bioactive GAs at the shoot apex by repressing the *GA20ox2* gene [204]. In response to inductive photoperiods, repression of SVP contributes to the increase of GAs at the shoot apex, promoting rapid induction of flowering. The ambient temperature and GA pathways are tightly linked (Figure 4, [205]).

Cytokinins (CK) were also proposed to affect flowering time as exogenous application of CKs can promote flowering in Arabidopsis. It has been shown that exogenous cytokinins promote flowering independently of *FT*, but through the transcriptional activation of its paralog *TSF* [189]. Cytokinins are perceived by membrane-located receptors called *A. THALIANA HISTIDINE KINASE2* (*AHK2*), *AHK3* and *AHK4* and are involved in many plant processes during plant development. Gain-of-function variants of *AHK2*, with enhanced cytokinin signaling, showed either early or late flowering [206]. Furthermore, it has been suggested that there is a cross-talk between cytokinins and GA, mediated by *SPINDLY* (*SPY*) [207].

In *B. napus*, genes encoding DELLA proteins and genes of the GA metabolism have been identified [208], however their role in flowering was not explored. During *B. napus* vernalization, the content of cytokinins increases significantly and reaches a maximum during reproductive transitions. Cis-Zeatin riboside accounted for ca. 87% to 89% of the total isoprenoid cytokinin content in control and vernalized plants, whilst isopentenyladenosine and cis-zeatin were the next most abundant cytokinins. In the post-vernalization period, endogenous cytokinin levels decreased, but remained significantly higher in the reproductive plants than in the vegetative controls. Changes in cytokinin accumulation during vernalization-induced reproductive development may suggest a possible role of CK in this process. [209].

In *B. rapa*, low-temperature treatment increases the GA content, and enhanced GA accumulation initiates floral bud differentiation [210]. Expression patterns of most genes involved in GA metabolism, particularly those of four genes including one *GA20ox* were consistent with observed GA levels [210].

In *B. oleracea*, treatment with bioactive GA_3_ and GA_4+7_ result in early curd formation in cauliflower and broccoli plants [184]. GA treatment induces bracting and stem elongation, but not flower initiation, when cauliflower and broccoli are at the IM or floral bud stage, respectively. As confirmation, treatment with GA does not show differences in the expression of *BoAP1-a*, *BoAP1-c*, *BoLFY* and *BoSOC1* in cauliflower plants at the IM stage. These results suggest that GA has an effect on vegetative-to-reproductive transition and another pathway is responsible for the IM-to-FM transition [184].

In *R. sativus*, two homologs for *GID1a*, one for *GID1b*, one for *GID1c* and three for *GA2ox* have been described. Before vernalization, expression level of one *RsGA2ox* homolog was upregulated in a late compared to early bolting line. Expression level of one homolog of *RsGID1a* was induced by vernalization treatment [104].

#### 4.6.2. Hormonal Pathway in Asteraceae

In *L. sativa*, plants treated with exogenous GA have enhanced levels of GA_3_ and GA_4_ in the leaves and flower early, with an enhanced effect in early flowering varieties [61,124]. Early flowering plants treated with CCC (a GA inhibitor) have reduced GA_3_, GA_4_ and IAA levels in the leaves and stem, are compact and do not bolt. Transferring plants from ambient (25/15°C) to higher temperature (35/25 °C) results in enhanced expression level of *LsGA2ox1*, *LsGA3ox1* with corresponding enhanced endogenous levels GA_8_ and GA_1_, respectively. Expression level of *LsGA20ox1* and corresponding endogenous level of GA_20_ was unaffected by the transfer to higher temperature. Therefore, it is suggested that *LsGA3ox1* might be responsible for enhanced bioactive GA_1_ levels in plants grown at higher temperatures [211]. With transcriptome analysis of a bolting resistant and sensitive line, Han et al. [61] have shown that *LsGA3ox1*, *LsGA20ox1*, *LsGA20ox2* and 28 out of 41 auxin-related genes were upregulated in leaves of a bolting sensitive line. Liu et al. [124] showed that heat treatment of bolting sensitive plants results in early bolting, enhanced GA_3_ and GA_4_ levels in the leaves, reduced IAA levels in the leaves and enhanced IAA levels in the stem. Transcriptomic analysis of a bolting sensitive line has shown that, out of 1443 and 1038 differentially up and down regulated genes, *LsGA20ox* was upregulated in leaves and a gibberellin-regulated family protein upregulated in the stem tip after heat treatment [124]. *L. sativa* plants overexpressing Arabidopsis *KNAT1*, a KNOTTED1-like homeobox (KNOX) transcription factor, show altered plant architecture and early flowering compared to control plants. Their striking leaf morphology phenotype was associated to a consistent increase in cytokinin content. Based on these results, correlation between temperature, GA levels and flowering time is suggested, together with a role of KNAT1 in flowering time, directly or indirectly, through cytokinins [212]. It has been proposed that the KNOX transcription factor *KNAT1* could regulate flowering by increasing cytokinin biosynthesis [212], and *ISOPENTENYL TRANSFERASE* (*IPT*) biosynthetic genes were shown to be downstream targets of KNOX transcription factors [213,214]. However, there is no direct evidence that the early flowering phenotype observed in *KNAT1* overexpressing lettuce plants depends on CK increase as KNOXs also control other hormonal and metabolic pathways, including GA biosynthesis (reviewed in [215]). Hence, the observed early flowering phenotype may depend on mis-regulation of as of yet unknown targets in the flowering time genetic network.

## 5. Quantitative Trait Loci (QTL)

The identification and functional characterization of genes controlling different pathways of flowering time has increased the knowledge about this complex trait. In parallel, the genetic basis of natural variation in flowering time has been investigated by quantitative trait loci (QTL) analysis. Salomé et al. [216] and Brachi et al. [217] have described the QTL mapping of 17 F2 populations and 13 RIL (recombinant inbred line) families in *A. thaliana*, which has led to the identification of many QTLs. Most of the QTLs are located in five genomic regions (region At1–5) and contain flowering time genes previously described in this review (Figure 4). All the five QTL regions described contain large-effect alleles [216,217]. Within the detected QTL regions, epistatic interaction between *FLC* and *FRI* alleles is highly associated with flowering time and could explain up to 70% of the variation [218,219]. Recently, a genome wide association map of flowering time, with nearly complete genotype information, was obtained taking advantage of the genomic sequencing and phenotype information from different environments (10 °C and 16 °C) of 1135 natural inbred lines of *Arabidopsis thaliana* [65]. The identified peaks from the genome wide association study (GWAS) contained *VIN3*, *FT*, *SVP*, *FLC* and *DOG1*, all previously linked to flowering time [81,123,164,220,221].

In *B. napus*, a mapping population made from a cross between Tapidor (winter type) and Ningyou7 (semi-winter type) is the most used for the identification of QTLs affecting flowering time [5,66,222,223,224,225]. Other analyses include different mapping populations or a broad set of accessions and inbred lines. Overall, phenotyping was performed in field trials in different locations and over multiple years, and flowering time was scored when 25% or 50% of the plants within a plot had an open flower. Many QTLs have been discovered in the different populations, with 23 genomic regions (Bn1-23) overlapping between at least two QTL analyses (Table 4). Of the flowering related genes within these genomic regions, *Bna.FRI.Xa* (region Bn5) is shown to have specific haplotypes overrepresented in either semi-winter or winter type plants [3,14]. Long et al. [222] have shown that genomic region Bn13 explains 50% of the variation in flowering time, is specific for spring environments and suggested that *Bna.FLC.A10* might control flowering time in non-vernalization environments. Later, Hou et al. [224] observed that one of the polymorphic sites upstream of *Bna.FLC.A10* is strongly associated with vernalization requirement of rapeseed. For *Bna.FT.A07b* (region Bn11), differential expression between types or treatments has been described [47], but no haplotype information is available so far.

In *B. rapa*, QTL analyses were performed on mapping populations mainly involving Yellow Sarson or a rapid cycling line 09A001. Phenotyping was scored based on flowering time (days to first open flower) or bolting time (days to first internode elongation). Of the detected QTLs in different populations, six genomic regions (Br1–6) were overlapping in at least two QTL analyses (Table 4). Of the flowering related genes within these genomic regions, *BrFLC2* (region Br2) is a major factor in determining flowering time. A 57 bp deletion on the exon4/intron4 border of *BrFLC2*, resulting in alternative splicing, is significantly associated with flowering time [91]. Zhang et al. [49] showed that a transposon insertion in exon 2 of *BrFT2* (region Br5) results in late flowering, and that there is a correlation between flowering time and different *BrFLC2* and *BrFT2* alleles. Plants with functional or non-functional alleles for both genes result in similar flowering time. However, a non-functional allele of either *BrFLC2* or *BrFT2* results in early or late flowering, respectively [49]. Besides the QTLs detected in multiple analysis, Xie et al. [153] has described one QTL (ChrA09:25634145.25774304), containing *BrGI* as a candidate gene responsible for circadian period determination. Two detected *BrGI* alleles (*BrGI^imb211^* and *BrGI^500^*) could complement the late flowering phenotype of the Arabidopsis *GI* null mutant, but plants with allele *BrGI^500^* showed a shorter circadian period and could not (fully) complement the response to red and blue light [153].

In *B. oleracea*, different mapping populations and commercial parents have been used for QTL analysis. Phenotyping was performed in the greenhouse and was scored as days to flowering or days to curd initiation (curd larger than 1 cm). Of the detected QTLs in different populations, six genomic regions (Bo1–6) were overlapping in at least two QTL analyses (Table 4). One of the candidate genes in region Bo1 is *BoFLC4* (Table 4). The two main alleles BoFLC4^E5^ and BoFLC4^E9^ both confer a requirement for vernalization, but differ with regard to their transcription regulation in response to temperature shifts, due to cis-regulatory differences [101]. One of the candidate genes in region Bo1 is *BoFRIa* (Table 4). Sequencing of *BoFRIa* from 55 accessions detected six different alleles with numerous substitutions and InDels. Expression of the two most abundant alleles from the *AtFRI* promoter prolonged the time to flower equally when overexpressed in *A. thaliana*, suggesting that the potential effect of these alleles on flowering time in *B. oleracea* may result from differences in their expression [90].

For *L. sativa*, a RIL population of cultivar *L. sativa cv. Salinas* (Crisphead) and Californian *L. serriola* unveiled two QTLs (Ls1 and Ls3) for days to flowering [226,227]. Furthermore, backcrossed lines selfed for one generation (BC1S1) from a cross between cultivar *L. sativa cv. Dynamite* (Butterhead) and a *L. serriola* uncovered four additional QTLs (Ls2, Ls4–6) [228]. A few flowering time related genes are located within QTL regions Ls1, Ls5 and Ls6. However, it remains an open question whether polymorphisms in these candidate flowering time related genes underlie the detected QTLs. Recently the *L. sativa* genome sequence [20] and RNA-seq data from 240 wild and cultivated lettuce accessions were realized, which will provide valuable tools to explore genetic variations contributing to flowering time and other traits in *L. sativa* [229].

In general, *FLC* and *FRI* seem to be overlapping in QTL analyses between different Brassica species. This provides more evidence that indeed these are key regulators of flowering time in many Brassicaceae. However, the specific genes and alleles responsible for the other QTLs remain unexplored. Identification of the causal genes and genetic variation for all QTLs would help to further understand the regulation of flowering time in the different crops. More QTL analyses have been performed for some species other than those discussed here. Even though these data are of great value, it is difficult to determine if the QTLs overlap with the reported QTLs, as reported positions cannot be related to the physical map (e.g., QTL analysis from [230,231]). For *L. sativa*, only two populations have been used for QTL analysis, both involving wild source *L. serriola*. It might be worthwhile exploring other wild sources such as *L. virosa* or *L. saligna* to expand the number of currently known QTLs. The availability of new genetic and genomics resources will consistently speed up genetic studies to unravel the key regulatory nodes of flowering time pathways in Asteraceae leafy crops.

## 6. Perspectives for Breeding Strategies

Knowledge about conservation and divergence of *A. thaliana* flowering time with its related crop species, and with more distant leafy crops within the Asteraceae family, is of great value to select candidate genes for the improvement of flowering time in commercial varieties. Introducing genetic variation in those candidate genes can be achieved by identifying novel alleles from wild relatives, the production of mutant populations or, when allowed, via a transgenic or genome editing approach.

### 6.1. Environmental Changes

Breeders aim to produce commercial varieties that are more robust and predictable in flowering time to adapt to climate change and new environments.

In cauliflower, exploring genetic variation in temperature-dependent flowering time genes such as *SVP*, *FLM* and *FLC* [3,101,107] would help in adjusting the vernalization and temperature sensitivity of plants for a predictable curd formation.

In lettuce, exploring genetic variation in the floral integrator genes *FT* and *SOC1* will help to understand the mechanism of heat-induced early flowering and can therefore be used to produce better tasting lettuce when grown at high temperatures. Different studies have described that silencing of either *LsFT* or *LsSOC1* results in late flowering and heat insensitive lettuce plants [36,59]. *LsSOC1* expression was enhanced in both heat-treated wild type and *LsFT* silenced lines, indicating that *LsSOC1* can induce bolting independent of *LsFT* upon heat treatment [36]. Heat shock elements (HSE1 and HSE2) are detected in the promoter of *LsSOC1* and two heat shock proteins (LsHsfA1e and LsHsfA4c) bind to these elements to induce flowering [36]. Genetic variation at the heat-responsive promoter elements of *SOC1* might selectively affect heat sensitivity rather than flowering time in general.

### 6.2. Yield Increase

Prolonged vegetative phase can increase yield in leafy crops that are harvested before the transition to the reproductive phase.

In radish, premature bolting under LD conditions reduces yield and quality of the harvested product. Delayed bolting is described for *RsGI* loss-of-function mutants in *R. sativus* [154], while silencing of *BoGI* in *B. oleracea* also resulted in delayed post-harvest leaf senescence. Based on the phenotype of the *B. oleracea* silencing line, it is worthwhile to test the effect of genetic variation in *RsGI* as added value of delayed leaf senescence together with delayed bolting.

Cold season during growth induces early bolting and decreases yield in root chicory. A *C. intybus* homolog of the Arabidopsis *FLC*, *CiFL1*, was characterized and seems conserved in the vernalization response [22]. However, it remains to be demonstrated that the high expression level of *CiFL1* in non-vernalized chicory plants is the cause of the absolute vernalization requirement for flowering. This indicates that more research about the vernalization response of chicory is required to achieve late bolting plants when grown at low temperature.

### 6.3. Genetic Resources

In the past centuries, domestication has led to the creating of edible vegetables from their wild relatives. During this domestication process, plants are selected for specific desirable traits, thereby losing some of the genetic variation in the current germplasm. As a result, some variation in flowering time genes, producing crops that are adapting to specific environments, are not present in our current breeding material. Exploring phenotypic and genotypic differences in closely related (wild) species, and introducing desired traits back into breeding material, will help create new varieties that are adapted to climate change and produce higher yield.

The Brassicaceae family contains both annual, biennial and perennial species, and spring, semi-winter and winter type plants within a species, indicating that this family varies greatly in flowering time response [3,86]. Within the family, different family members are closely related and can be crossed through interspecific crosses, making it easier to introduce new genetics. Hybridization between *R. sativus* and Brassica species *B. napus*, *B. rapa* and *B. oleracea*, and between *D. tenuifolia* and *B. rapa* has been proven to be successful even though the number of successful hybridizations might be rather low [244]. Schiessl et al. [35] have described the amount of copies of 35 flowering time regulatory genes and their genetic variation between *B. rapa* and *B. oleracea*. This genetic information could be used as a basis to look for candidate genes to follow in an interspecific cross. As an example, Shea et al. [245] have developed late flowering *B. rapa* plants by replacing the *BrFLC2* genomic region with a 6.5 Mb region containing *BoFLC2* from *B. oleracea*. As many of the flowering pathways are conserved within the Brassicaceae family, it is worthwhile to explore introgression of flowering time genes from Brassica species into *R. sativus* or *D. tenuifolia* to alter flowering time.

Introducing genetic variation in flowering time genes from wild material into cultivated lettuce and chicory is possible [246,247], however, the flowering pathway is largely undiscovered in these species. Recently, high quality transcriptomes of both *C. intybus* and *C. endivia* were obtained by de novo assembly using RNA of several organs and Illumina HiSeq2000 technology [248,249], paving the way to the identification of flowering time transcripts in *Cichorium* spp. More research is required before specific candidate genes can be selected to introduce from wild material into breeding lines.

### 6.4. Speeding up Breeding

From a breeding perspective, introducing genetic variation from wild relatives or mutant populations into a new variety will take up to years. Speeding up this breeding process, using early flowering plants to grow more generations in one year would be of added value for the breeding companies. Similar to adapting plants to climate change or increasing yield, generating early flowering plants is possible by the use of genetic variation in flowering time genes. Besides exploring the genetics of wild material, it has also been shown for lettuce that screening mutant populations are a great source to discover plants with an altered flowering time. In Brassica spp., rapid-cycling lines and RIL populations have been obtained [86], which can be used to speed up breeding and for rapid analysis of QTL.

In Arabidopsis, winter type plants that require vernalization contain functional alleles for both *FLC* and *FRI*, while summer type plants lack a functional allele for either *FLC* or *FRI* [70,71,72,73]. With this system, early bolting parental lines can be created, while the F1 hybrids are late flowering. As an example, by producing a female line containing an *FLC* knock-out and a male line with FRI knock-out. The parents do not require vernalization to initiate flowering, as both parents lack a functional allele for either *FLC* or *FRI*. In the F1 hybrid, both genes are heterozygous, resulting in winter type plants that do require vernalization.

Different articles have shown that treatment with bioactive GAs can induce early flowering in both bolting sensitive and bolting resistance lettuce lines [61,124]. The benefit of GA application is that it will speed up the breeding process, when this is desired, but will not have a negative influence on flowering time during crop production.

## 7. Conclusions

Overall, most flowering time pathways seem to be genetically conserved between Brassicaceae and Asteraceae families, paving the way for exploitation of the fundamental knowledge acquired in the Brassica model species Arabidopsis to closely or more distantly related vegetable crops. This is highlighted in Figure 5, which represents a simplified model of the main regulatory genes shown to have a function in the various species within the Brassicaceae or Asteraceae family. However, a comprehensive comparison of the different flowering time pathways between Brassicaceae and Asteraceae is impaired by the poor knowledge available about molecular biology and gene function in *D. tenuifolia*, *L. sativa* and *C. intybus*. Fundamental biology studies in crop species to identify casual genes of advantageous traits is advisable to apply candidate gene approaches for successful breeding strategies. An increasing number of tools for molecular marker assisted breeding is expected to come in the near future from genomic and transcriptomic studies. With the rapid development of sequencing technology, whole genome sequences assembly and resequencing from crop plants is becoming routine, enabling genome-wide investigations into fundamental genetic pathways that underlie important agricultural traits. In addition, generating a pan-genome, capturing the genomic diversity of ecotypes, geographical isolates, and domesticated crop varieties, will make comparative approaches and association studies possible to identify the genetic components of adaptive and domestication traits. Increasing “omics” information (e.g., genomics, transcriptomics, metabolomics, SNP-omics) will enable systems biology approaches to understand complex traits, such as flowering time, and identify hub/master gene regulators for the so-called “smart” or “precision breeding,” which aims to develop new varieties more precisely and rapidly.

## Figures and Tables

**Figure 1 plants-07-00111-f001:**
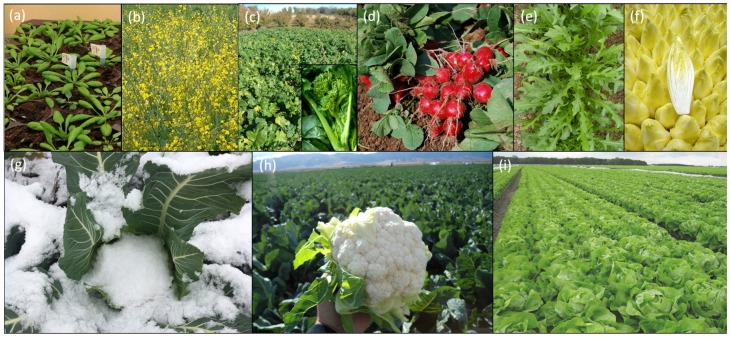
Plant species considered in this review. (**a**) *Arabidopsis thaliana*, (**b**) *Brassica napus* (rapeseed), (**c**) *Brassica rapa* (turnip), (**d**) *Raphanus sativus* (radish), (**e**) *Diplotaxis tenuifolia* (wild rocket), (**f**) *Cichorium intybus* (chicory), (**g**) *Brassica oleracea* (cauliflower) in winter, (**h**) *Brassica oleracea* (cauliflower) in summer and (**i**) *Lactuca sativa* (lettuce).

**Figure 2 plants-07-00111-f002:**
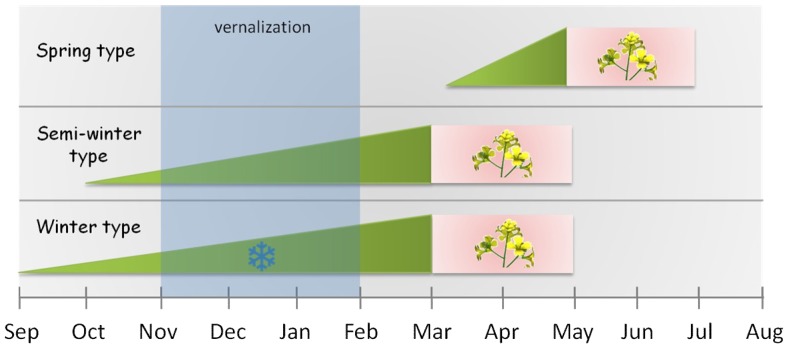
Schematic representation of the life cycles of annual Brassica species. Green triangles represent vegetative growth, pink boxes plant flowering. Periods of cold required for vernalization are indicated by a blue box. Frost symbols indicate frost hardiness in winter types that does not occur in semi-winter plants.

**Figure 3 plants-07-00111-f003:**
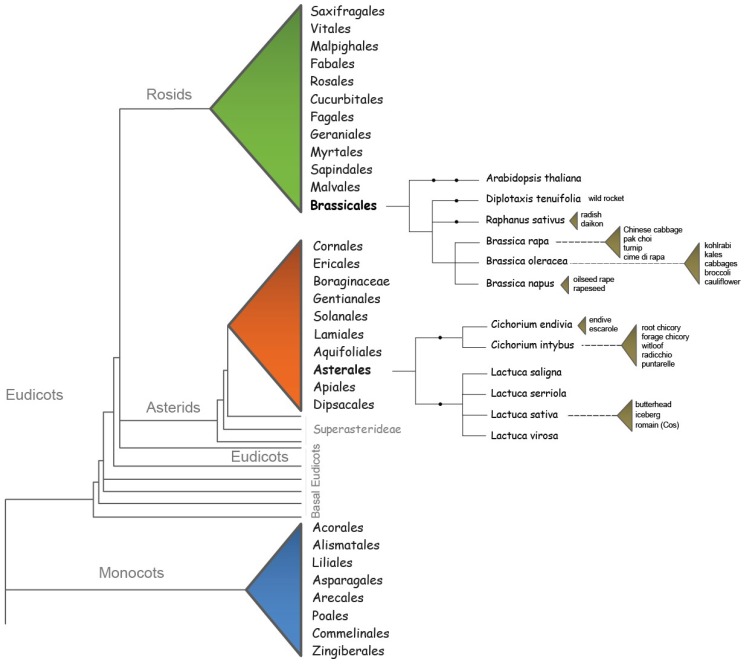
Schematic phylogenetic tree of eudicot and monocot species. Plant families are indicated for the main phylogenetic groups (http://science.kennesaw.edu/jmcneal7/plantsys/index.html). Phylogenetic relationships within Brassicaceae and Asteraceae species of interest were obtained using phyloT, a phylogenetic tree generator based on NCBI taxonomy (https://phylot.biobyte.de/). Cultivated crops for the different plant species are shown.

**Figure 4 plants-07-00111-f004:**
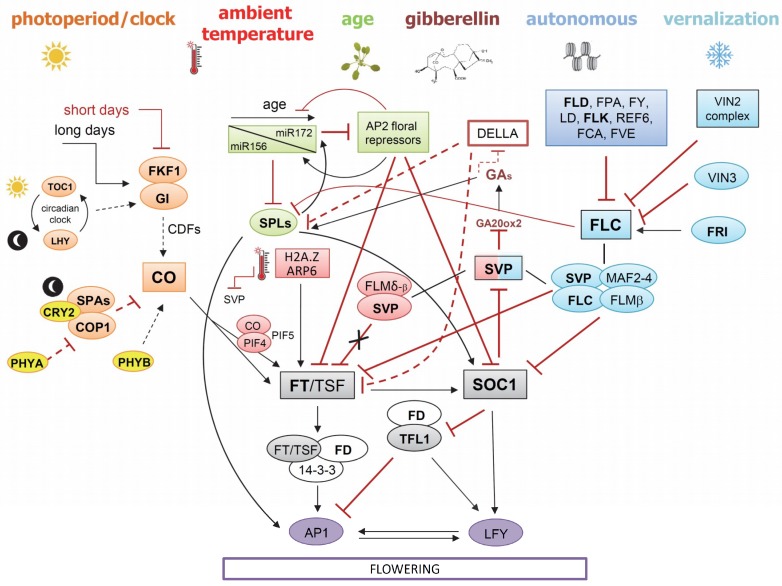
Main flowering time pathways acting in *Arabidopsis thaliana*: Photoperiod (orange and yellow), ambient temperature (red), age (green), gibberellins (brown), autonomous (sky blue), vernalization (light blue). Grey boxes represent the main floral integrators FT/TSF and SOC1. The two main genes conferring inflorescence meristem identity, *AP1* and *LFY*, are indicated in purple. Squared boxes indicate genes having a pivotal role in the specific pathway. Boxes with rounded corners represent several genes or complexes. Solid and dotted lines indicate either direct or indirect regulation, black arrows and red T-ends indicate positive or negative regulation, respectively. The cartoon represents only the main regulatory genes in the different pathways, whereas the complete flowering time network, involving more than 300 genes, is available at the WikiPathways Web Site [33].

**Figure 5 plants-07-00111-f005:**
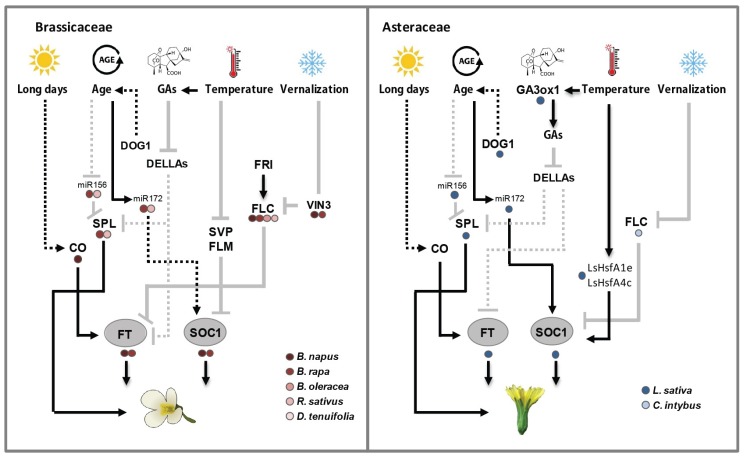
Simplified model of the main regulatory genes and flowering pathways acting in various crops within the Brassicaceae and Asteraceae family. Grey and black lines represent repression and induction, respectively, dotted lines indicate indirect regulation.

**Table 1 plants-07-00111-t001:** Glossary of main terms as used in the review.

Term	Definition
**flowering time**	the switch from plant vegetative growth to reproductive development
**bolting**	rapid elongation of the inflorescence/flowering stem
**annuals**	plants that complete their entire life cycle from seed to flower within one year and are characterized by short vegetative phase
**biennials**	plants which require two years to complete their life cycle,
**perennials**	plants that survive for several years and restrict the duration of reproduction by cycling between vegetative growth and flowering; perennials are characterized by prolonged vegetative phase that can last from a few weeks to several years
**shoot apical meristem (SAM)**	population of cells located at the tip of the shoot axis that produce lateral organs, stem tissue and regenerates itself
**inflorescence meristem (IM)**	a meristem that underwent transition from vegetative to reproductive fate and can produce floral meristems
**floral meristem (FM)**	group of cells responsible for the formation of floral organs
**facultative photoperiod**	plants that flower faster under a particular photoperiod but will eventually flower under all photoperiods (also called “quantitative”)
**obligate photoperiod**	plants that flower only under a particular photoperiod (also called “qualitative”)
**long days**	day length more than about 12 h, usually 16 h light and 8 h dark periods
**short days**	day length less than about 12 h, usually 8 h light and 16 h dark periods
**Double Haploid (DH)**	chromosome doubling of haploid cells to produce genetically homozygous plants
**genome-wide association study (GWAS)**	observational study of a genome-wide set of genetic variants in different individuals that occur more frequently in correlation with a specific trait, identifying inherited genetic variants associated with a trait
**homolog**	a gene related to a second gene by descent from a common ancestral DNA sequence
**ortholog**	genes in different species that evolved from a common ancestral gene by speciation; normally, orthologs retain the same function in the course of evolution
**paralog**	genes related by duplication within a genome that may evolve new functions
**maturity**	the state of being fully developed or full grown
**uniformity**	a state or condition of the plant in which everything is regular, homogeneous, or unvarying
**predictable**	always behaving or occurring in the way expected
**robust**	is a characteristic of being strong that, when transposed into a system, it refers to the ability of tolerating perturbations and remain effective
**QTL**	(or Quantitative Trait Locus), is a locus (section of DNA) which correlates with variation of a quantitative trait in the phenotype of a population of organisms
**vernalization**	cold treatment needed to get many perennials to flower; usually the minimum period is six to twelve weeks at 4 °C
**spring types**	plants which flower early without vernalization
**winter types**	plants which have an obligate requirement for prolonged periods of cold temperatures
**semi-winter types**	plants which require mild vernalization and lack frost hardiness

**Table 2 plants-07-00111-t002:** Brief description of the plant species.

Species	Chr.	Life Span	Vernalization	Types	Breeding Goal	Day Length ^1^	Ref.
***A. thaliana***	2n = 10	Annual/biennial	Yes/no	Spring/semi-winter/winter	none	facultative LD	[12]
***B. napus***	2n = 38 (AACC)	Annual/biennial	Yes/no	Spring/semi-winter/winter	flowering time adaptation	LD	[13,14]
***B. rapa***	2n = 20 (AA)	Annual/biennial	Yes/no	Spring/semi-winter/winter	Late bolting	LD	[15]
***B. oleracea***	2n = 18 (CC)	Annual/biennial	Yes/no	Spring/semi-winter/winter	Predictable harvest time	LD	[16]
***R. sativus***	2n = 18	Annual	No		Late bolting	facultative LD	[17,18]
***D. tenuifolia***	2n = 22	Annual	No		Late bolting	LD	[19]
***L. sativa***	2n = 18	Annual	No		Heat resistance	facultative LD	[20]
***C. intybus***	2n = 18	Biennial/perennial	Yes		Resistance to bolting	LD	[21,22]

^1^ All species flower early under long-day (LD, 16 h/8 h of light/dark) conditions.

**Table 3 plants-07-00111-t003:** Flowering time genes in Brassica species.

Pathway	Gene	Arabidopsis	*B. napus* gene ID	*B. napus* Chr position	*B. rapa* gene ID	*B. rapa* Chr position	*B. oleracea* gene ID	*B. oleracea* Chr position
Floral integrators	FT	AT1G65480	GSBRNA2T00090951001 Bna.FT.A02	A02:6375936.6379058	Bra022475 BrFT1	A02:8551268.8553758		
			GSBRNA2T00030311001 Bna.FT.C02	C02:996695.998788			Bol045330	Scaffold000001_P2: 1990327.1992083
	TSF	AT4G20370	GSBRNA2T00124448001 Bna.FT.A07	A07:18855196.18857952	Bra004117 BrFT2	A07:20213069.20215397		
			GSBRNA2T00146560001	A07:22787807.22790354	Bra015710	A07:24515213.24516895		
			GSBRNA2T00077948001	C02:20907503.20909228			Bol039209	C02:19450855.19452577
			GSBRNA2T00113342001	C04:12435074.12437644			Bol017639	C04:17148775.17151658
			GSBRNA2T00067517001 Bna.FT.C06	C06:28552966.28555216			Bol012573	C07:9349005.9351279
			GSBRNA2T00050890001	Cnn:48285424.48286397			Bol027595	C07:1423408.1425153
	TFL1	AT5G03840	GSBRNA2T00136426001	A10:16767409.16768474	Bra009508	A10:15774055.15775120		
			GSBRNA2T00119620001	Ann:609805.611005	Bra028815	A02:545667.546787		
			GSBRNA2T00078727001	C02:1320757.1321835			Bol005471	C02:1447642.1448756
			GSBRNA2T00134290001	C03:673349.674628			Bol015337	C03:438359.439413
			GSBRNA2T00073025001	Cnn:9572005.9573076			Bol010027	C09:39511589.39512660
					Bra005783	A03:603455.604516		
	SOC1	AT2G45660	GSBRNA2T00011646001	A03:901877.905188 BnSOC1-A3	Bra000393	A03:10918286.10920672		
			GSBRNA2T00063263001	A04:18732428.18735897	Bra039324	A04:18723546.18725960		
			GSBRNA2T00116723001	A05:2627051.2630394	Bra004928	A05:2530305.2532747		
			GSBRNA2T00037309001	C04:48074887.48078345			Bol021742	C04:40413670.40414880
			GSBRNA2T00083011001	C04:867297.870707			Bol030200	C04:2998426.2999594
			GSBRNA2T00029970001	Cnn:35198162.35204681			Bol029556	C03:13421127.13422327
Vernalization	FLC	AT5G10140	GSBRNA2T00143535001 Bna.FLC.A02	A02:134362.138212	Bra028599 BrFLC2	A02:1524995.1528254		
			GSBRNA2T00129741001	A03:1360971.1364359	Bra006051 BrFLC3	A03:1764912.1767856		
			GSBRNA2T00142187001	A03:6240056.6245305	Bra022771 BrFLC5	A03:6971946.6976797		
			GSBRNA2T00135921001	A10:14998617.15003197	Bra009055 BrFLC1	A10:13856133.13860473		
			GSBRNA2T00068991001 Bna.FLC.C02	C02:208562.212139			Bol024642 BoFLC4	C02:2720826.2721596
							BoFLC2	C02:2722189.2724345
			GSBRNA2T00134620001	C03:2001058.2004665			Bol008758 BoFLC3	C03:1890867.1893743
			GSBRNA2T00024568001	C03:8403312.8410062			BoFLC5	C03:49708405.49709316
			GSBRNA2T00016124001	C09:46345350.46350092			Bol043693 BoFLC1	C09:37175182.37179020
			GSBRNA2T00016119001	C09:46366645.46371180				
	FRI	AT4G00650	GSBRNA2T00066686001 Bna.FRI.Xa	A03:6053113.6055294	Bra029192 BrFRIa	A03:6784863.6787013		
			GSBRNA2T00120967001 Bna.FRI.Xb	A10:4019556.4021675	Bra035723 BrFRIb	A10:4133444.4134764		
			GSBRNA2T00052682001 Bna.FRI.Xd	C03:8149599.8151810			Bol028107 BoFRIa	C03:7962008.7964180
			GSBRNA2T00152364001 Bna.FRI.Xc	C09:29041826.29043953			Bol004294BoFRIb	Scaffold000327:204688.206816
Ambient temperature	SVP	AT2G22540	GSBRNA2T00032884001	A04:10961147.10963402	Bra030228	A04:10192172.10194736		
			GSBRNA2T00078179001	A09:29590705.29594744	Bra038511	A09:33434743.33437921		
			GSBRNA2T00149752001	C04:36478652.36481951			Bol031759	Scaffold000053:1406474.1408404
			GSBRNA2T00127429001	C08:32995398.32998881	Bol044741	C08:35213085.35214818		
Photoperiod	CO	AT5G15840	GSBRNA2T00135488001	A10:13358777.13360064	Bra008669	A10:12117648.12118929		
			GSBRNA2T00035272001	C09:43745679.43747139			Bol030488	C09:33143053.33144339
	GI	AT1G22770	GSBRNA2T00015763001	A09:22588149.22593013	Bra024536	A09:25756404.25760934		
			GSBRNA2T00119480001	C05:11778931.11784461			Bol023541	Scaffold000099_P1: 794479.799157
Age	SPL3	AT2G33810	GSBRNA2T00064576001	A04:15462653.15463366	Bra021880	A04:15123762.15124274		
			GSBRNA2T00095270001	A05:5425249.5426076	Bra005470	A05:5668800.5669314		
			GSBRNA2T00132295001	C03:9629272.9630113			Bol036997	C06:40526300.40526809
			GSBRNA2T00020688001	C04:44354526.44355241			Bol037895	C04:35540992.35541501
			GSBRNA2T00038835001	Cnn:4854484.4855112			Bol027299	C04:20510435.20510961
	SPL9	AT2G42200	GSBRNA2T00123166001	A04:17845227.17847617	Bra016891	A04:17839490.17841541		
			GSBRNA2T00132740001	A05:1443071.1445187	Bra004674	A05:1325605.1327587		
			GSBRNA2T00010840001	C04:1886612.1888780			Bol004847	C04:966922.968875
			GSBRNA2T00084688001	C04:46904649.46905101				
			GSBRNA2T00084692001	C04:46915351.46917939			Bol002678	Scaffold000379:152205.154297
					Bra015085	A07:5833985.5836119		
	SPL15	AT3G57920	GSBRNA2T00034335001	A04:154881.156614	Bra014599	A04:1684031.1685273		
			GSBRNA2T00098900001	A07:14658857.14660105	Bra003305	A07:15783674.15784920		
			GSBRNA2T00087887001	C04:25001142.25003655			Bol011022	C04:9176952.9178238
			GSBRNA2T00105779001	C06:19172101.19173360			Bol007052	C07:28887260.28888520
Gibberellin	GA20OX1	AT4G25420			Bra013890	A01:8279446.8280885		
					Bra019165	A03:25974634.25976038		
							Bol039527	C01:11622628.11624071
							Bol042237	C06:43862307.43863764
							Bol041615	Scaffold000009_P1: 317473.317667
	GA20OX2	AT5G51810	GSBRNA2T00036929001	A02:5851980.5853392	Bra022565	A02:7878457.7879856		
			GSBRNA2T00110217001	A10:6243369.6244766	Bra028277	A10:4556457.4557854		
			GSBRNA2T00153037001	C02:11109632.11111035			Bol045266	Scaffold000001_P2: 739593.740993
			GSBRNA2T00108688001	C09:30358041.30359437			Bol029404	C09:18392061.18393124
			GSBRNA2T00108686001	C09:30368578.30369971				
			GSBRNA2T00025818001	Cnn:77727013.77728046				
	GA20OX3	AT5G07200			Bra028706	A02:1065763.1067251		
					Bra005927	A03:1251910.1253082		
					Bra010064	A06:14066314.14068214		
					Bra009285	A10:14501966.14503398		
							Bol024532	C02:2131323.2133012
							Bol008872	C03:1262233.1263937
							Bol043862	C09:38109369.38110750
							Bol041616	Scaffold000009_P1: 331428.332030
							Bol024814	Scaffold000091:1192479.1195919
	GA20OX4	AT1G60980	GSBRNA2T00070537001	A09:7950969.7952898	Bra027106	A09:8952409.8954196		
			GSBRNA2T00043758001	Ann:21491303.21492435	Bra039251	Scaffold000162: 175188.176593		
			GSBRNA2T00043759001	Ann:21493033.21494525	Bra031467	A01:17039613.17041240		
			GSBRNA2T00080857001	Cnn:11274341.11276251			Bol014320	C03:52855996.52857792
			GSBRNA2T00028142001	Cnn:34189212.34190700			Bol044153	Scaffold000003_P1: 2530105.2531509
							Bol007374	Scaffold000262:280424.282063
	GA20OX5	AT1G44090	GSBRNA2T00097054001	A08:969550.970698	Bra014019	A08:4525615.4527790		
			GSBRNA2T00060328001	C08:6199021.6199389				
			GSBRNA2T00013490001	Cnn:72340763.72341689			Bol021441	C07:30202821.30203904

Note: Gene symbols, gene names and position in the chromosome of the paralogs for *B. napus*, *B. rapa* and *B. oleracea*, and corresponding *A. thaliana* genes, are shown for each pathway. Black text means the same chromosome in all Brassica species, grey text means best *B. napus* hit from *B. rapa* or *B. oleracea* protein sequence; brown text means no direct homolog available in the list with *A. thaliana* syntheny (http://brassicadb.org/brad/searchAll.php).

**Table 4 plants-07-00111-t004:** Flowering-time related QTL regions for *A.* thaliana, *B.* napus, *B.* rapa, *B.* oleracea and *L. sativa* with candidate flowering-time genes within these QTL regions.

QTL Region	Species	Region ^1^	Candidate Genes	References
At1	*A. thaliana*	Chr1:24500000-29000000	*FT*, *FKF1*, *AP1*, *FLM*	[216,217]
At2	*A. thaliana*	Chr4:300000-1900000	***FRI***	[216,217]
At3	*A. thaliana*	Chr4:8000000-12000000	*VRN2*, *TSF*, *GA2ox*	[216,217]
At4	*A. thaliana*	Chr5:2700000-8100000	***FLC***, *CO*, *TFL2*	[216,217]
At5	*A. thaliana*	Chr5:21500000-26000000	*VIN3*, *PRR3*, *TOE2*, *LFY*, *CDF1*, *MAF2-5*	[216,217]
Bn1	*B. napus*	chrA02:114931.1575498	*Bna.FLC.A2*, *CO-like*, *RVE1*	[232,233]
Bn2	*B. napus*	chrA02:1575449.4330821	*AP2-like*, *TOE2*, *PRR3*	[225,232,233]
Bn3	*B. napus*	chrA02:5233136.8233310	*GA20ox*, *Bna.FT.A02*	[223,225,232,233,234,235]
Bn4	*B. napus*	chrA02:8776742.9248051		[5,222,234,236]
Bn5	*B. napus*	chrA03:5046910.6515058	***Bna.FRI.Xa***, *SPL13*, *CBF1*, *Bna.FLC.A03b*	[66,222,234,236]
Bn6	*B. napus*	chrA03:18872718.20131639	*AP2-like*, *FUL*, *TOC1*	[223,232,233,236]
Bn7	*B. napus*	chrA04:257040.4734286	*AP2-like*	[233,234,236]
Bn8	*B. napus*	chrA04:7743947.10942653		[233,234]
Bn9	*B. napus*	chrA04:11898475.13460703	*CO-like*, *ELF3*	[234,236]
Bn10	*B. napus*	chrA06:23330530.23617143		[232,236]
Bn11	*B. napus*	chrA07:14463578.18916565	*SPL15, AP2-like*, *GID1*, *AP1*, ***Bna.FT.A07b***	[232,233,234,236]
Bn12	*B. napus*	chrA10:9835903.10695100	*PRR3*, *TOE2*, *AP2-like*	[222,234]
Bn13	*B. napus*	chrA10:13375104.15191366	***Bna.FLC.A10***	[66,222,224,233,234]
Bn14	*B. napus*	chrC01:27417076.34893173	*FRI-like*, *VRN1*	[232,233]
Bn15	*B. napus*	chrC02:6956919.13653054	*GA20ox*, *SPL*	[222,232,234]
Bn16	*B. napus*	chrC02:22287455.22560553		[222,234]
Bn17	*B. napus*	chrC02:44366336.45788246	*FUL*, *MAF2*, *MAF3*	[225,232]
Bn18	*B. napus*	chrC03:58161161.58296560		[233,234]
Bn19	*B. napus*	chrC04:40003810.41181656		[222,234]
Bn20	*B. napus*	chrC06:21784608.29654361	*ELF4*, *AP1*	[225,232,233,236]
Bn21	*B. napus*	chrC07:26989258.31787256	*SEP4*	[225,232,234]
Bn22	*B. napus*	chrC09:39312343.43429210	*SPL7*, *AP2-like*, *TFL2*, *RVE*	[234,236]
Bn23	*B. napus*	chrC09:45206288.47504024	*Bna.FLC.C09b*, *Ga20ox*	[225,232]
Br1	*B. rapa*	A01:81263.3282650	*AP2-like*	[237,238]
Br2	*B. rapa*	A02:1244721.4284193	***BrFLC2***, *AP2-*like, *CO-*like, *SPL7*	[49,92,237,238,239]
Br3	*B. rapa*	A03:14357780.27239372	*CO-*like, *AP2-*like, *GA2ox*, *AGL24*	[237,238,239]
Br4	*B. rapa*	A06:13769411.18840509	*LFY*, *GA20ox*, *CDF1*, *FLM*, *MAF4*, *VIN3-*like, *CO-*like,	[238,240]
Br5	*B. rapa*	A07:12545242.20240840	*AP2-*like, *SPL15*, *ELF4-*like, *AP1*, ***BrFT2***	[49,238,240]
Br6	*B. rapa*	A10:12936259.13856133	*BrFLC1*	[237,238,241]
Bo1	*B. oleracea*	C02:900000.2900000	*GRF6*, ***BoFLC4***	[101,242]
Bo2	*B. oleracea*	C03:1800000.20000000	*BoFLC3*, *SOC1*, ***BoFRIa***, *ELF4*, *GA20ox*	[100,242,243]
Bo3	*B. oleracea*	C04:10726862.16070000	*TOE2*	[100,243]
Bo4	*B. oleracea*	C04:32446947.35540000		[100,244]
Bo5	*B. oleracea*	C06:2396965.6360269	*TOE1*, *VIN3*	[242,243]
Bo6	*B. oleracea*	C06:22550000.32446947		[100,243]
Ls1	*L. sativa*	LG2:163353056.165477161	*CDF1*, *CO*, *FLC*, *PRR5*, *VRN1*	[226,227]
Ls2	*L. sativa*	LG6:140450832.140481276		[228]
Ls3	*L. sativa*	LG7:158780460.159063877		[226,227]
Ls4	*L. sativa*	LG7:172306237.193636147		[228]
Ls5	*L. sativa*	LG8:25874939.47456612	*PRR3*, *PRR5*, *PRR7*, *PRR9*	[228]
Ls6	*L. sativa*	LG8:63537238.76202393	*FKF1*	[228]

^1^ Regions on genomes of *A. thaliana* (Tair10), *B. napus* (Brassica_napus_v4.1.chromosomes), *B. rapa* (Brapa_genome_sequence_v1.5), *B. oleracea* (B. oleracea var. capitate V1.0) and *L. sativa* (lettuce genome V8.1). ^2^ For Brassica species, only QTLs detected in more than one study, encompassing different mapping populations and/or varieties, are shown. For lettuce, only two populations have been used for QTL mapping, Flowering time genes with described allelic variation are highlighted in bold.

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
