# Peer review of "Translating Flowering Time from Arabidopsis thaliana to Brassicaceae and Asteraceae Crop Species"

_plants, 2018, doi:10.3390/plants7040111_

Round 1

Reviewer 1 Report

“Translating flowering time from A. thaliana to Brassicaceae and Asteraceae crop species” is a comprehensive review for the current status on flowering time regulating network in two crop families. It will serve as a valuable resource and reference for the continuous exploration.

I only have few minor suggestions:

1.       Fig.2. The authors tried to used difference colors and probably shapes to depict the pathways, but sometimes it introduced another layer of confusion, such as FKF1 and GI, both have the same orange background and oval shape, but GI has red font while FKF1 is in black. Remove “considered in this review” from subtitle 2

2.       Is “BnaA3.FRI” (line 329) the same gene as “BnFRI-A3” (line 332)?

3.       Line 543, the statement is not clear.

4.       Line 936, what is the “bigger effect”?

5.       Few typos, such as missed parentheses at line 66, “Diplotaxis. tenuifolia” at line 112, “2n-=18” at line 132, some information missed for Ref 45, and “long day” or “long-day”.

Author Response

Reviewer 1. Minor suggestions:

First of all, we want to thank the Reviewer for his valuable suggestions

1.       Fig.2. The authors tried to used difference colors and probably shapes to depict the pathways, but sometimes it introduced another layer of confusion, such as FKF1 and GI, both have the same orange background and oval shape, but GI has red font while FKF1 is in black. Remove “considered in this review” from subtitle 2

Authors response: In the new version of Fig.2 (that is now Fig.3) we used black font for all the genes in the pathway and explained in the legend the use of squared boxes for the main pathway regulators and that of smooth angles boxes to indicate multiple genes or protein complexes. We removed “considered in this review” from subtitle 2.

2.       Is “BnaA3.FRI” (line 329) the same gene as “BnFRI-A3” (line 332)?

Authors response: Yes, it was the same gene but we realized that different authors use their own personal nomenclature for Brassica genes, and this creates a lot of confusion. Hence, we decided to use the nomenclature from Schiessl et al, Frontiers in Plant Science 2017, all over the manuscript, following the format “Bna/Bra/Bol.FT.C06” to indicate the specific paralogs and  “Bn/Br/BoFT” when classes of homologous genes were discussed.

3.       Line 543, the statement is not clear.

Authors response: We changed the sentence into “D. tenuifolia plants flower later under SD compared to LD conditions, with 50 and 20 days to flowering, respectively”

4.     Line 936, what is the “bigger effect”?

Authors response: We agree, the meaning was not clear at all, we changed the whole sentence and cancelled “bigger effect”

5.       Few typos, such as missed parentheses at line 66, “Diplotaxis. tenuifolia” at line 112, “2n-=18” at line 132, some information missed for Ref 45, and “long day” or “long-day”.

Authors response: We revised the whole manuscript for minor typos mistakes

Reviewer 2 Report

In this review, the authors discuss the flowering time and related pathways, comparing Arabidopsis thaliana with the crops from Brassicaceae and Asteraceae.

Overall, this review paper is very informative, well-written with a good flow. My main concern is that it requires more figures to be able to follow the information given in the text better. For example, a phylogenetic tree of the species compared could be provided. Also, while describing the gene expression of paralogs, providing a visual representation would be helpful.

Author Response

Responses to Reviewer 2

First of all, we want to thank the Reviewer for valuable suggestions

Overall, this review paper is very informative, well-written with a good flow. My main concern is that it requires more figures to be able to follow the information given in the text better. For example, a phylogenetic tree of the species compared could be provided. Also, while describing the gene expression of paralogs, providing a visual representation would be helpful.

Authors response: We do agree with you and added two additional figures and two novel tables: novel Figure 2 is the schematic representation of the life cycles of annual Brassica species; novel Figure 3 represents the phylogenetic tree of eudicot and monocot species, the phylogenetic relationships of the considered species within Brassicaceae and Asteraceae and indication of the main cultivated crops for each species. A glossary table of main terms used in the review was added as actual Table 1, as suggested by Reviewer 3, and a new Table with all the Brassica gene names for flowering time genes, chromosomal location and gene ID for B. napus, B. rapa and B. oleracea was added as Table 3.

Reviewer 3 Report

Review of: Translating flowering time from A. thaliana to Brassicaceae and Asteraceae crop species. Leijten et al. submission to Plants

At its best, this manuscript succinctly and exhaustively reviews flowering time network knowledge across species and in doing so creates new expectations for which gene/pathway targets are likely to hold across species and thus be useful breeding targets. There is much to applaud here. This is a timely topic and the scope, detail, and clarity of writing are often very good. However, the sheer amount and complexity of the information also create major challenges for clarity and organization. The figures and tables actually do a good job of this, but I think there are substantive opportunities for improvement in the synthesis and clarity of the text that might be best addressed with structural changes. Throughout there is also a need for improvement in clarity of terminology (a glossary is needed) particularly in regards to categorizing life-cycles, varieties, and environmental sensitivities.

I would urge the authors to carefully consider who their intended audience is and how that audience will actually use this paper as a resource. As written, I suspect if a reader doesn’t have a very strong a priori background in the Arabidopsis flowering time pathway they are quickly going to get lost in the 13 pages describing not only the A. thaliana genes in each pathway but the pathways in each of the other species as well. In my opinion, most of the synthetic and big picture information and context for readers is in the last paragraph of each pathway section (e.g lines 252-258 line 552-560) and in the last 4 pages of the review. There are lots of ways to go about solving this issue structurally and I am certainly not going to mandate anything, but one possible solution is to consider flipping the structure so the big picture comes first and then the nitty-gritty details are last. I actually don’t think this would require much rewriting. I’ve outlined that idea below. Another idea is to at least flip the order within sections—summarize the big picture and take home messages about conservation first and then go into the details.

Set up the problem: Environments change temporally and spatially -> yield issues for crops including ones related to A. thaliana -> if we could get crops to flower when we wanted them to we could 1) max yields based on env. 2) synchronize flowering for harvest, and 3) speed up breeding, etc -> Lots of genetic resources are coming in line for the species we are highlighting 1) natural genetic variation 2) QTLs and GWAS panels, 3) tools for genetic manipulation -> if the flowering time genes in A.thaliana were conserved in these relatives they would provide targets for manipulation.

Big picture of FT network: Brief overview of each A. thaliana flowering time pathway (your current first paragraph of each section & summarized in Figure 1)+ your nice take home paragraph from each section that summarizes the overall pattern of “conservation and divergence” in the pathway (e.g lines 252-258 and summarized in current Figure 3) + a statement about how targeting that particular portion of the pathway would alter flowering and lead accomplish a breeding goal.

Detailed comparative analysis of all the specific genes: THEN have the 13 pages of nice details about each pathway that readers can dip into based on their specific interests once they have the big picture.

While the sections about genes and functions tend to be clearly written and well supported, statements about the adaptive value of flowering and the relationship between altering specific environmental sensitivities and flowering time and yield are not as well supported. For instance, the introductory paragraph (lines 46-56) makes many broad claims without any supporting citations.

New work is showing that it is the absence of warmth rather than the presence of cold that is necessary for vernalization (Hepworth 2018, DOI: 10.1038/s41467-018-03065-7). Similarly, there is lots of complexity and controversy in the ambient temperature sensing and pathway literature. Perhaps it is important to point out the areas of the flowering time pathway in A. thaliana that are well-established vs. under active revision?

Section 2 (Flowering requirements….lines 79-135) reads like an expanded version of Table 1. Perhaps it would be more effective to synthesize this for readers?

An example rewrite:  The four species most closely related to A. thaliana also share similar life cycles (annuals and biennials). All species flower faster flowering in response to longer photoperiods and warmer temperatures. They also all have multiple cultivar types (winter, semi-winter, spring) that differ in whether cold exposure (vernalization) is required prior to flowering….. Despite these similarities, breeding goals for each species differs….).

Table 1 does not define the categories abb. If all species are LD could you include the information about obligate vs. facultative instead of just listing LD for all of them? Do you know the critical photoperiods for these species?

The language at times can be imprecise and lack directionality at other times it very clear. If the point of this paper is to translate information for thinking about breeding, perhaps it is important to be explicit what the consequences of changing a gene are for flowering. Does it make plants flower later or earlier? Does it make plants insensitive to a cue? What is the effect on yield? I give an example of each below.

I found lines 170-177 to be very clear: “Overexpression of either FT or TSF results in early flowering, a mutation in FT results in late flowering under long day (LD) conditions and a mutation in TSF shows a greater effect under short (SD) conditions [33]. A more distantly related homolog of FT, TERMINAL FLOWER 1 (TFL1) with 59% amino acid identity, acts antagonistically to FT. Plants overexpressing TFL1 are late flowering with an extended first inflorescence phase, during which they form cauline leaves and branches [34].”

If found these less clear in regard to temperature changes. Are these changes heat waves, daily fluctuations, loss of vernalization?

Lines 844-846: In cauliflower (B. oleracea), changes in temperature lead to uneven curd formation and therefore less predictable harvest times.

Lines 97: temperature induced yield loss

Perhaps this is wrong, but I thought a benefit of genomic prediction was that one did not need to to know underlying genes? If that is the case, how important is it to know all the underlying pathways?

The meaning of the following sentences was not immediately obvious to me:

Lines 324-326: It has been shown in other species that gene clusters with functionally related genes might be maintained by selection pressure to enable adaptation to extremely diverse environments [80,81].

Lines 528-530: It was suggested that such rearrangements may represent a necessary co-adaptation of the photoperiodic pathway to the strong vernalization requirement in winter or swede inbred lines [142].—also I don’t know what “winter or swede” means?

Lines 554-555: Major diversity can be observed more between long-day plants and short-day plants rather than between eudicots and monocots

Tiny comments:

Figure 1 black and red are indistinguishable in black and white

Lines 892: Edible instead of eatable

Line 129: live stock instead of life stock

Similar small typos and tense issues occur throughout

A quick list of potential glossary terms:

Facultative vs. quantitative vs. obligate photoperiod response

Winter, semi-winter, spring varieties (also summer)--based on vern req?

Annual, biennial, perennial--- based on length of vegetative phase?

Bolting & Flowering

Vernalization

QTL & GWAS candidates

Double haploid

Uniformity, robust, predictable 

Author Response

Reviewer 3: major structural changes

First of all, we want to thank the Reviewer for valuable suggestions

I think there are substantive opportunities for improvement in the synthesis and clarity of the text that might be best addressed with structural changes. Throughout there is also a need for improvement in clarity of terminology (a glossary is needed) particularly in regards to categorizing life-cycles, varieties, and environmental sensitivities. I would urge the authors to carefully consider who their intended audience is and how that audience will actually use this paper as a resource.

Authors response: We deeply thank you for the time you dedicated to our manuscript to give us so many great suggestions to improve it: we did address the structural changes accordingly (see more details below) and prepared a glossary Table (Table 1 in the revised manuscript) with terms from fundamental biology as well as from breeding field: our intended audience is both the basic science community and the breeders, so we prepared the glossary Table keeping this in mind.

As written, I suspect if a reader doesn’t have a very strong a priori background in the Arabidopsis flowering time pathway they are quickly going to get lost in the 13 pages describing not only the A. thaliana genes in each pathway but the pathways in each of the other species as well. In my opinion, most of the synthetic and big picture information and context for readers is in the last paragraph of each pathway section (e.g lines 252-258 line 552-560) and in the last 4 pages of the review. There are lots of ways to go about solving this issue structurally and I am certainly not going to mandate anything, but one possible solution is to consider flipping the structure so the big picture comes first and then the nitty-gritty details are last. I actually don’t think this would require much rewriting. I’ve outlined that idea below. Another idea is to at least flip the order within sections—summarize the big picture and take home messages about conservation first and then go into the details.

Authors response: We agree that most of the synthetic and big information is in the last paragraph of each pathway section and in the last 4 pages of the review. Hence, we reassembled the manuscript flipping the structure as suggested: the first Sections became 1. Introduction (revised as suggested) – 2. Flowering requirements of Brassicaceae and Asteraceae species (that was also integrated with an additional Figure showing the annual Brassica life cycles, actual Figure 2) – 3. Breeding goals (describing the main breeding challenges, in part coming from the previous “Breeding strategies” Section at the end of the manuscript) – 4. Conserved and divergent flowering time genes in Brassicaceae and Asteraceae, which contains all the flowering pathway information restructured as sub and subsections (see below) – 5. Quantitative Trait Loci (QTL) – 6. Perspectives for breeding strategies – 7. Conclusions

Set up the problem: Environments change temporally and spatially -> yield issues for crops including ones related to A. thaliana -> if we could get crops to flower when we wanted them to we could 1) max yields based on env. 2) synchronize flowering for harvest, and 3) speed up breeding, etc -> Lots of genetic resources are coming in line for the species we are highlighting 1) natural genetic variation 2) QTLs and GWAS panels, 3) tools for genetic manipulation -> if the flowering time genes in A.thaliana were conserved in these relatives they would provide targets for manipulation. 

Authors response: We reorganized the Introduction according to your suggestions

Big picture of FT network: Brief overview of each A. thaliana flowering time pathway (your current first paragraph of each section & summarized in Figure 1)+ your nice take home paragraph from each section that summarizes the overall pattern of “conservation and divergence” in the pathway (e.g lines 252-258 and summarized in current Figure 3) + a statement about how targeting that particular portion of the pathway would alter flowering and lead accomplish a breeding goal. 

Detailed comparative analysis of all the specific genes: THEN have the 13 pages of nice details about each pathway that readers can dip into based on their specific interests once they have the big picture.

Authors response: We restructured each pathway section as suggested: i) a brief overview of the pathway with general information from Arabidopsis, a take home message about conservation and divergence and a final statement on how targeting the pathway; ii)  then, two subsubsections dedicated to detailed description of the pathway in Brassicaceae and in Asteraceae

While the sections about genes and functions tend to be clearly written and well supported, statements about the adaptive value of flowering and the relationship between altering specific environmental sensitivities and flowering time and yield are not as well supported. For instance, the introductory paragraph (lines 46-56) makes many broad claims without any supporting citations.

Authors response: We rearranged that part and insert citations when needed

New work is showing that it is the absence of warmth rather than the presence of cold that is necessary for vernalization (Hepworth 2018, DOI: 10.1038/s41467-018-03065-7). Similarly, there is lots of complexity and controversy in the ambient temperature sensing and pathway literature. Perhaps it is important to point out the areas of the flowering time pathway in A. thaliana that are well-established vs. under active revision? 

Authors response: We cited the new work from Hepworth et al. and indicated what pathways are still controversial in the pathway overview part

Section 2 (Flowering requirements….lines 79-135) reads like an expanded version of Table 1. Perhaps it would be more effective to synthesize this for readers? An example rewrite:  The four species most closely related to A. thaliana also share similar life cycles (annuals and biennials). All species flower faster flowering in response to longer photoperiods and warmer temperatures. They also all have multiple cultivar types (winter, semi-winter, spring) that differ in whether cold exposure (vernalization) is required prior to flowering….. Despite these similarities, breeding goals for each species differs….). 

Authors response: The flowering requirement part was synthesized following your nice example, thanks

Table 1 does not define the categories abb. If all species are LD could you include the information about obligate vs. facultative instead of just listing LD for all of them? Do you know the critical photoperiods for these species?

Authors response: The categories “obligate vs. facultative” were inserted in the Table

The language at times can be imprecise and lack directionality at other times it very clear. If the point of this paper is to translate information for thinking about breeding, perhaps it is important to be explicit what the consequences of changing a gene are for flowering. Does it make plants flower later or earlier? Does it make plants insensitive to a cue? What is the effect on yield? I give an example of each below. 

Authors response: Information for breeding were inserted at the end of each pathway overview

I found lines 170-177 to be very clear: “Overexpression of either FT or TSF results in early flowering, a mutation in FT results in late flowering under long day (LD) conditions and a mutation in TSF shows a greater effect under short (SD) conditions [33]. A more distantly related homolog of FT, TERMINAL FLOWER 1 (TFL1) with 59% amino acid identity, acts antagonistically to FT. Plants overexpressing TFL1 are late flowering with an extended first inflorescence phase, during which they form cauline leaves and branches [34].”If found these less clear in regard to temperature changes. Are these changes heat waves, daily fluctuations, loss of vernalization?

Lines 844-846: In cauliflower (B. oleracea), changes in temperature lead to uneven curd formation and therefore less predictable harvest times. Lines 97: temperature induced yield loss

 Authors response: We better clarified cauliflower response to temperature: “In cauliflower, slight deviations from optimal growth temperature, either lower or higher, lead to uneven curd  formation”. Same for line 97, we clarified with this sentence “Extremely late bolting is a major breeding goal in this crop as unexpectedly low temperatures can induce flowering and so yield loss”

Perhaps this is wrong, but I thought a benefit of genomic prediction was that one did not need to to know underlying genes? If that is the case, how important is it to know all the underlying pathways? 

Authors response: It is important for a candidate gene approach aiming to discover mutations to fine-tune a specific trait without affecting others. These mutations are difficult to discover in standard experiments of GWAS

The meaning of the following sentences was not immediately obvious to me: 

Lines 324-326: It has been shown in other species that gene clusters with functionally related genes might be maintained by selection pressure to enable adaptation to extremely diverse environments [80,81]. 

Authors response: The sentence was changed to “… gene clusters with functionally related genes might be maintained by selection pressure to enable adaptation to extremely diverse environments in a similar manner as the cold-responsive cluster FLC-FRI-CBF1”

Lines 528-530: It was suggested that such rearrangements may represent a necessary co-adaptation of the photoperiodic pathway to the strong vernalization requirement in winter or swede inbred lines [142].—also I don’t know what “winter or swede” means? 

Authors response: We eliminated “swede” as they are specific B. napus subspecies that are not relevant for the review and were not discussed.

Lines 554-555: Major diversity can be observed more between long-day plants and short-day plants rather than between eudicots and monocots

Authors response: The sentence was changed to “Photoperiod and circadian rhythm are involved in many processes of adaptive response to environmental conditions, including flowering time. Their molecular mechanisms are widely conserved amongst plant species to such an extent that mechanisms of photoperiod measurement are more diverse between long-day and short-day plants than between eudicots and monocots” and inserted the appropriate reference

Tiny comments: 

Figure 1 black and red are indistinguishable in black and white

Lines 892: Edible instead of eatable

Line 129: live stock instead of life stock

 Similar small typos and tense issues occur throughout

Authors response: Tiny comments were addressed

A quick list of potential glossary terms:

Facultative vs. quantitative vs. obligate photoperiod response

Winter, semi-winter, spring varieties (also summer)--based on vern req?

Annual, biennial, perennial--- based on length of vegetative phase?

Bolting & Flowering

Vernalization

QTL & GWAS candidates

Double haploid

Uniformity, robust, predictable 

Authors response: Thank you for the suggestions, we used those to prepare our glossary Table

Round 2

Reviewer 3 Report

The revised manuscript is substantively improved in terms of both content and clarity and in my view is basically ready to publish. I believe this review will be well-received and will make a substantive contribution to the field. I am attaching a pdf where I highlighted phrases that I found be vague or awkwardly worded.

A couple of tiny comments:

1) Perhaps the monocot portion of the phylogeny Figure is unnecessary if space is an issue.

2) Consider changing the text colors in Table 3 so they can't be confused with the pathway color coding

3) Perhaps it would be clearer to refer to the annual and perennial categories as defined by the length of the vegetative period of the life cycle.

4) The definitions of orthologs and paralogs could be clearer 

Author Response

The revised manuscript is substantively improved in terms of both content and clarity and in my view is basically ready to publish. I believe this review will be well-received and will make a substantive contribution to the field. I am attaching a pdf where I highlighted phrases that I found be vague or awkwardly worded.

Dear Reviewer,

we deeply appreciated your help in improving the structure of the manuscript and in highlighting mistakes and phrases that were not clear, and thank you for the accuracy and time dedicated to us. We changed all the points you highlighted in the file to improve readers’ understanding.

A couple of tiny comments:

1) Perhaps the monocot portion of the phylogeny Figure is unnecessary if space is an issue.

As space is not an issue, we kept the monocot portion of the phylogeny in Figure 3.

2) Consider changing the text colors in Table 3 so they can't be confused with the pathway color coding

We changed the text colors in Table 3 and adjusted the pathway color coding to better fit with Figure 4

3) Perhaps it would be clearer to refer to the annual and perennial categories as defined by the length of the vegetative period of the life cycle.

We changed the glossary table accordingly to highlight differences in the length of  vegetative phase in annuals and perennials

4) The definitions of orthologs and paralogs could be clearer 

We changed the definitions of orthologs and paralogs in the glossary table to make them clearer
